# Seasonal and inter-annual variability of $CO_2$ fluxes in southern Africa seen by GOSAT

Eva-Marie Metz[1], Sanam Noreen Vardag[1,2], Sourish Basu[3,4], Martin Jung[5], André Butz[1,2,6]

[1]Institute of Environmental Physics, Heidelberg University, Heidelberg, 69120, Germany
[2]Heidelberg Center for the Environment (HCE), Heidelberg University, Heidelberg, 69120, Germany.
[3]Goddard Space Flight Center, NASA, Greenbelt, MD 20771, USA
[4]Earth System Science Interdisciplinary Center, University of Maryland, College Park, MD 20740, USA
[5]Max Planck Institute for Biogeochemistry, Jena, 07745, Germany
[6]Interdisciplinary Center for Scientific Computing (IWR), Heidelberg University, Heidelberg, 69120, Germany

*Correspondence to*: Eva-Marie Metz (eva-marie.metz@iup.uni-heidelberg.de)

**Abstract.** The inter-annual variability of the global carbon sink is heavily influenced by semi-arid regions. Southern hemispheric Africa has large semi-arid and arid regions. However, there is only a sparse coverage of in situ $CO_2$ measurements on the southern hemisphere. This leads to uncertainties in measurement-based carbon flux estimates for these regions. Also, dynamic global vegetation models (DGVMs) show large inconsistencies in semi-arid regions. Satellite $CO_2$ measurements offer a spatially extensive and independent source of information about the southern African carbon cycle.

We examine Greenhouse Gases Observing Satellite (GOSAT) $CO_2$ concentration measurements from 2009 to 2018 in southern Africa. We infer $CO_2$ land-atmosphere fluxes which are consistent with the GOSAT measurements using the atmospheric inversion system TM5-4DVar. We find systematic differences between atmospheric inversions performed on satellite observations versus inversions that assimilate only in situ measurements. This suggests limited measurement information content in the latter. We use the GOSAT based fluxes and additionally Solar Induced Fluorescence (SIF), a proxy for photosynthesis, as atmospheric constraints to select DGVMs of the TRENDYv9 ensemble which show compatible fluxes. The selected DGVMs allow for studying the vegetation processes driving the southern African carbon cycle. Doing so, our satellite-based process analyses pinpoint photosynthetic uptake in the southern grasslands to be the main driver of the inter-annual variability of the southern African carbon fluxes, agreeing with former studies based on vegetation models alone. We find that the seasonal cycle, however, is substantially influenced by enhanced soil respiration due to soil rewetting at the beginning of the rainy season. The latter result emphasizes the importance of correctly representing the response of semi-arid ecosystems to soil rewetting in DGVMs.

## 1 Introduction

The terrestrial carbon sink currently takes up nearly one third of human made greenhouse gases and thereby mitigates climate change (Friedlingstein et al., 2023). The amount of $CO_2$ taken up by global ecosystems varies substantially from year to year. This inter-annual variability (IAV) reflects the response of ecosystem carbon uptake to varying climate conditions

such as temperature or precipitation fluctuations (Zeng et al., 2005; Zhang et al., 2018; Piao et al., 2020). Current vegetation models struggle in accurately reproducing IAV of the terrestrial carbon sink and an imbalance exists between the modelled and measured total global sink estimates (Friedlingstein et al., 2023). The imbalance is even stronger when examining

carbon fluxes on smaller spatial scales (Bastos et al., 2020) and implies that there is still an insufficient understanding of the terrestrial processes driving land carbon exchange. A better understanding is needed to improve climate models and climate change predictions (Steiner et al., 2020).

Semi-arid regions contribute substantially to the IAV of the global terrestrial carbon sink. In these regions, precipitation and temperature fluctuations heavily impact the IAV of carbon fluxes (Poulter et al., 2014; Ahlström et al., 2015). Africa has

large areas of semi-arid and arid ecosystems (Williams et al., 2007) and contributes substantially to the global IAV (Williams et al., 2007; Valentini et al., 2014; Pan et al., 2020). However, in situ $CO_2$ measurements in Africa are very sparse leading to large uncertainties in carbon flux estimates from atmospheric inversions and machine learning approaches (Valentini et al., 2014; Ernst et al., 2024). Dynamic Global Vegetation Models (DGVMs), also, show large inconsistencies amongst each other and tend to underestimate the inter-annual $CO_2$ flux variability in semi-arid regions (MacBean et al.,

45    2021).

Satellite $CO_2$ concentration measurements, for example from the Greenhouse Gases Observing Satellite (GOSAT) measuring $CO_2$ concentrations since 2009 or the Orbiting Carbon Observatory-2 (OCO-2) launched in 2014, have much denser coverage compared to in situ measurements. Previous studies found systematic differences between satellite- and in situ measurement-based $CO_2$ concentrations and fluxes in southern Africa (Mengistu and Mengistu Tsidu, 2020; Byrne et al.,

2023). Byrne et al. (2023) attribute these differences mainly to the sparse coverage of in situ $CO_2$ measurements. The studies emphasize the potential of satellite-based atmospheric inversions to provide additional information and therefore more robust estimates of the carbon fluxes in southern hemispheric Africa, which then enable research about processes driving the $CO_2$ exchange. Metz et al. (2023) demonstrate the potential of combining satellite-based $CO_2$ flux estimates with DGVMs in Australia to decipher soil respiration processes driving the Australian terrestrial $CO_2$ exchange on continental scale.

Here, we investigate the decadal dataset of GOSAT $CO_2$ concentrations over southern hemispheric Africa from 2009 to 2018. We run a global inversion with GOSAT and in situ measurements to infer GOSAT satellite-based $CO_2$ exchange between land and atmosphere and compare the results to those based on in situ measurements alone, to FLUXCOM products, and to the TRENDYv9 ensemble of DGVMs. By selecting a subset of DGVMs which match the satellite-based carbon fluxes, we analyze the underlying processes driving the IAV and seasonal variability of the southern African carbon

cycle.

## 2 Data and Methods

### 2.1 Study region

Our study region spans southern hemispheric Africa southwards of 10° S including Madagascar (see Fig. 1). This region agrees with the region selection in Mengistu and Mengistu Tsidu (2020) taking into account the different climatic conditions in the African continent. Northwards of the study region, Africa is influenced by the low pressure system of the inter tropical convergence zone leading to a tropical wet regime. In the southern Africa, high pressure cells lead to dry conditions and cause the existence of the Kalahari Desert (Mengistu and Mengistu Tsidu, 2020). Even though total annual precipitation is decreasing southwards, the whole region experiences distinct wet and dry seasons and is influenced strongly by IAV of precipitation (Fan et al., 2015; Valentini et al., 2015). The study region is mainly covered by (woody) savannas, grassland, and shrubland (see Fig. 1).

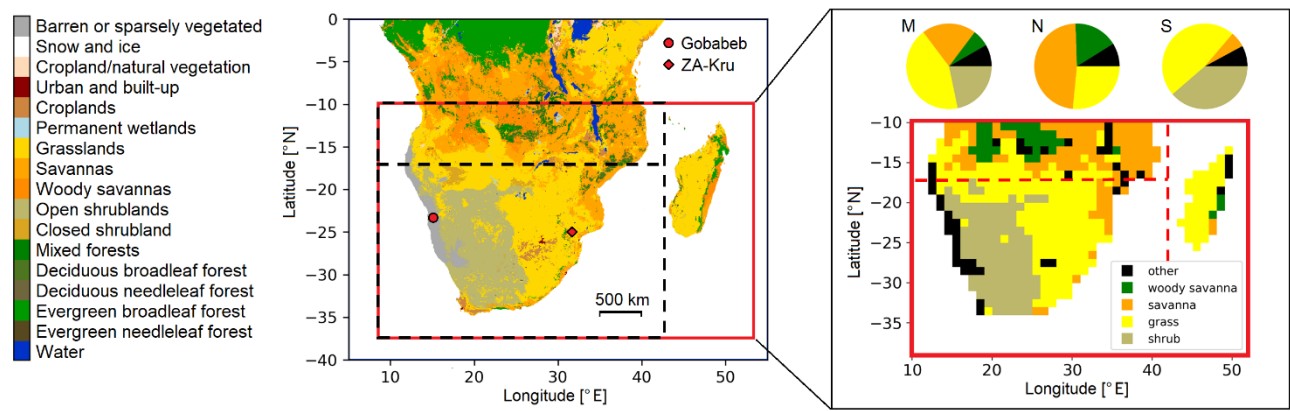

Figure 1: Study region southern Africa. The land cover in the study region is given based on MODIS (MCD12C1) data (Friedl and Sulla-Menashe, 2022). Additionally, the main region used for the analyses is depicted as a red box. In the inlet map on the right side, the land cover is aggregated in larger land cover classes and on a 1°x1° spatial resolution, which is used for most of the analyzed data. The main region, thereby, comprises 547 grid cells. The dashed boxes show the subdivision into a northern and southern region. Madagascar is part of the main region, but it is excluded in the subdivision. The pie charts depict the share of the different land cover classes in the main study region (M), the northern subregion (N), and the southern subregion (S). The locations of the COCCON measurement site Gobabeb (Frey et al., 2021; Dubravica et al., 2021) and the flux tower in Kruger National Park (Archibald et al., 2009) are given as red circle and diamond.

The vegetation is mostly water limited in its growth (Williams et al., 2008) and exposed to large seasonal fires. The fire season starts in May in the western part of southern hemispheric Africa and spreads eastwards to reach southern Africa in September (Edwards et al., 2006). Fires on the whole African continent are the largest contributor to and account for more than half of the global fire carbon emissions (van Marle et al., 2017; Shi et al., 2015; Valentini et al., 2014). They reduce the African carbon sink significantly (Lasslop et al., 2020). We subdivide the study region in a northern, savanna dominated region and a southern grass- and shrubland region separated at 17°S, excluding Madagascar.

## 2.2 Total column $CO_2$ measurements

For our analyses we use column-averaged dry-air mole fractions of $CO_2$ ($XCO_2$, in the following referred to as $CO_2$ concentrations) measured by the Greenhouse Gases Observing Satellite (GOSAT) over land in our study region. GOSAT was launched in 2009 and has a sub-satellite field of view of 10.5 km radius with a sparse sampling grid. We use GOSAT

CO₂ concentration data generated by applying the RemoTeC radiative transfer and retrieval algorithm version 2.4.0 (Butz, 2022) as used in Metz et al. (2023). The retrieval version covers the period 04/2009–06/2019 and is based on the preceding RemoTeCv2.3.8 as used in Detmers et al. (2015). The major updates between versions 2.3.8 and 2.4.0 are stricter quality filtering in the latter and updated ancillary input data, especially for the prior gas concentrations used. Moreover, GOSAT CO₂ concentration data generated by the NASA Atmospheric CO₂ Observation from Space (ACOS) algorithm version 9 (Lite), available for the period 04/2009-06/2020, is used (Taylor et al., 2022). In the following the datasets are called GOSAT/RemoTeC and GOSAT/ACOS (see Table A1 for more information about the datasets and nomenclature used in this study). GOSAT/ACOS single measurements have a precision of 1.5 ppm and a mean bias of 0.2 ppm in validation against TCCON (Taylor et al., 2022). GOSAT/RemoTeC was found to have a similar precision of 1.9 ppm (Buchwitz et al., 2017) and by construction a mean bias of 0 ppm in comparison to TCCON after bias correction. GOSAT/RemoTeC was found to have a regional and seasonal systematic error of 0.6 ppm and 0.5 ppm respectively (Buchwitz et al., 2017).

For evaluation purposes, Land-Glint and Land-Nadir (LGLN) XCO₂ data (version 11.1r) measured by the Orbiting Carbon Observatory-2 (OCO-2) satellite is used (OCO-2/OCO-3 Science Team, 2022; Jacobs et al., 2024). OCO-2 was launched in 2014 and has a sub-satellite field of view of 1.3 km x 2.3 km. Furthermore, Collaborative Carbon Column Observing Network (COCCON) XCO₂ data of the Gobabeb station (Namibia, Frey et al., 2021; Dubravica et al., 2021) is taken for comparison. COCCON stations measure XCO₂ using a sun-viewing ground-based Fourier transform infrared spectrometer (Frey et al., 2019). We use the full dataset of COCCON measurements i.e. we do not apply further filtering or co-sampling to GOSAT, as there are too few coinciding GOSAT measurements.

For examining the seasonal variability of CO₂ concentrations in the study region, the global background trend is subtracted from the total CO₂ measurements to obtain detrended CO₂ concentrations. For this, we assume a yearly linear increase of global atmospheric CO₂ and use the annual mean CO₂ growth rate (GR) published by the National Oceanic and Atmospheric Administration (NOAA). The growth rates are based on globally averaged CO₂ concentration measurements of marine surface sites (NOAA, 2024) and their calculation is further described in Taylor et al. (2023, Figure A3) and Pandey et al. (2024). The following equation describes the used background trend:

$$BG_{y,m} = BG_0 + \sum_{i=2009}^{y-1}(GR_i) + \frac{m}{12}GR_y \,. \tag{1}$$

Thereby, the increase of the CO₂ concentrations in the previous years from 2009 onwards is described by the second part in the equation. The increase within the previous months in the respective year is given by the third part. Both are added to an overall offset BG₀ in 2009. This offset is estimated so that the mean of the detrended CO₂ concentrations over the whole time period is zero.

**2.3 Fluxes**

**2.3.1 Top-down**

Carbon fluxes can be obtained by assimilating measured $CO_2$ atmospheric concentrations in an atmospheric inversion. Atmospheric inversions typically build on Bayesian optimization i.e. they optimize forward transported $CO_2$ emissions such that these agree best with the observations within measurement and model uncertainties, while at the same time not deviating from the prior within given prior uncertainties. For our study, we use three in situ $CO_2$ measurement based atmospheric

inversions: the TM5 four-dimensional variational inversion system (TM5-4DVar, Basu et al., 2013), NOAA's modelling and assimilation system CarbonTracker (CT2022, Peters et al., 2007; Jacobson et al., 2023), and Copernicus Atmosphere Monitoring Service (CAMS, Chevallier et al., 2005; Chevallier et al., 2010; Chevallier et al., 2019). The models estimate global $CO_2$ fluxes based on a set of in situ $CO_2$ measurements from global monitoring networks (Masarie et al., 2014). The models use different prior datasets. For example, for the biogenic $CO_2$ fluxes, TM5-4DVar and CarbonTracker build on

different implementations of the Carnegie-Ames-Stanford Approach (Randerson et al., 1996) as further described in Metz et al. (2023), Weir et al. (2021), and Jacobson et al. (2023), while CAMS uses biogenic fluxes of the ORCHIDEE model (Chevallier et al., 2019). Furthermore, the inversion systems use different transport models and inversion techniques. While TM5-4DVar and CarbonTracker use the transport model TM5, CAMS uses the LMDZ global atmospheric transport model. TM5-4DVar and CAMS make use of a four-dimensional variational data assimilation, while CarbonTracker uses an

ensemble Kalman filter. All three models use ECMWF ERA5 data as meteorological drivers. The output resolution is monthly 3°x2° for TM5-4DVar and CarbonTracker2022 and monthly 3.7°×1.81° for CAMS (see Table A1 for more details). The ensemble of the three models is referred to as in-situ-only inversions in the following, TM5-4DVar based on in situ measurements is called TM5-4DVar/IS.

In addition to in situ measurements, satellite $CO_2$ concentration measurements can be assimilated by atmospheric inversions.

To this end, we use the model TM5-4DVar and assimilate GOSAT $CO_2$ concentration measurements over land and ocean together with the in situ measurements. We use the individual total $CO_2$ concentration measurements, i.e. we do not apply any detrending or spatiotemporal averaging. Detrending and spatiotemporal averaging is only applied for visualization purposes to show the variability in the monthly $CO_2$ concentrations (Section 3.1). Depending on the specific GOSAT dataset used, we refer to these fluxes in the following as TM5-4DVar/RemoTeC+IS, TM5-4DVar/ACOS+IS, or when using the

mean of both TM5-4DVar/GOSAT+IS. More details about the TM5-4DVar settings can be found in Metz et al. (2023). For comparison we also draw on data of the OCO-2 Model Intercomparison Project (MIP) (Byrne et al., 2023) for the years 2015 to 2018. Within the MIP, atmospheric inversions estimate carbon fluxes by assimilating OCO-2 satellite $XCO_2$ observations together with in situ data. All MIP inversion models use the same fossil fuel emission dataset but differ in the chosen datasets for all other prior fluxes (Byrne et al., 2023). We specifically make use of the LNLGIS (assimilation of OCO-2

LNLG observations together with in situ measurements) and the IS (assimilation of in situ measurements only) experiment in the following referred to as MIP/OCO-2+IS and MIP/IS, respectively. Like Byrne et al. (2023), we exclude the MIP

model LoFI as it uses a non-traditional inversion scheme differing from the MIP protocol. MIP/OCO+IS and MIP/IS provide fluxes with monthly 1°x1° resolution.

All inversions optimize for biogenic and oceanic fluxes but impose anthropogenic fossil fuel emissions and fire emissions. The sum of (imposed) fire and biogenic fluxes yields our net biome productivity (NBP) estimates. In this study, positive fluxes denote a release of $CO_2$ from land into the atmosphere. All fluxes are regridded to monthly 1°x1° fluxes before performing the region selection.

By transporting the posterior fluxes after the optimization, atmospheric inversions can model posterior concentration fields, which can be interpolated to the time and location of the satellite measurements for comparison. This so-called cosampling is used to eliminate sampling errors when comparing modelled concentrations to satellite measurements. We use the modelled and co-sampled posterior concentrations of the in-situ-only inversions introduced at the beginning of this section.

### 2.3.2 Bottom-up

We compare the top-down $CO_2$ fluxes to bottom-up flux datasets from DGVMs as collected by version 9 of the intercomparison project "trends and drivers of the regional-scale sources and sinks of carbon dioxide (TRENDY, Le Quéré et al., 2013). The project was established to support the annual global carbon budget estimation conducted by the Global Carbon Project (e.g. Friedlingstein et al., 2020). These TRENDY models give vegetation $CO_2$ fluxes simulated using a harmonized set of meteorological input data and $CO_2$ concentrations (Le Quéré et al., 2013; Friedlingstein et al., 2020). We use the NBP, gross primary productivity (GPP), autotrophic respiration (RA), and heterotrophic respiration (RH) of 18 DGVMs (see Table A1). We thereby use the following definition:

$$NBP = NEE + fire + fluc = TER - GPP + fire + fluc = RH - NPP + fire + fluc, \qquad (2)$$

with the total ecosystem respiration (TER) calculated as sum of RA and RH, the fire emissions (fire), the land-use change fluxes (fluc), and the net primary productivity (NPP) calculated as GPP – RA. Most of the TRENDY models provide NBP fluxes directly. In the case of the models CABLE-POP and DLEM, NBP is calculated as RH-NPP, as both models do not provide fire and land-use change fluxes. The spatial resolutions of the model output differ (see Table A1). Therefore, we aggregate fluxes on a monthly 1°x1° grid before applying the region selection.

Additionally, we use the FLUXCOM net ecosystem exchange (NEE) product version 1 (setup RS_V006) as described in Jung et al. (2020). FLUXCOM uses machine learning models and meteorological data to upscale eddy covariance tower $CO_2$ flux measurements to global scale (Tramontana et al., 2016; Jung et al., 2020). To obtain an NBP estimate, we combine the NEE fluxes with fire $CO_2$ emissions provided by the Global Fire Emission Database (GFED, van der Werf et al., 2017). FLUXCOM and GFED are provided as 0.08°x0.08° 8-day fluxes and 0.25°x0.25° daily fluxes, respectively, and are aggregated on a monthly 1°x1° grid before applying the region selection.

## 2.4 Other datasets

To investigate the climatic conditions influencing the carbon fluxes, we use temperature, upper layer soil moisture, and precipitation datasets of the European Centre for Medium Range Weather Forecasts (ECMWF) ERA5-land data product (Muñoz Sabater et al., 2019; Muñoz Sabater et al., 2021) with monthly resolution on a 0.25°x0.25° spatial grid. ERA5 datasets are aggregated on a 1°x1° grid before performing the region selection. Furthermore, we use Solar Induced fluorescence (SIF) measurements by the GOME-2 satellite from 2009 to 01/2018 (Joiner et al., 2023). SIF is considered proportional to GPP on monthly time scale and biome resolution (Sun et al., 2018; Joiner et al., 2018; Pierrat et al., 2022; Zhang et al., 2016a; Zhang et al., 2016b). It can therefore be used as a proxy for $CO_2$ uptake by photosynthesis (Li et al., 2018).

## 3 Results

### 3.1 Monthly $CO_2$ concentrations by atmospheric inversions

To access the seasonal and inter-annual dynamics in southern Africa, we detrend the monthly mean $CO_2$ concentrations following Eq. (1) (see Data and Methods Sec. 2.2). The remaining $CO_2$ enhancements for the study region are shown in Fig. 2. The GOSAT measured $CO_2$ enhancements reveal a clear seasonal cycle with minimum concentration in the first and maximum concentrations in the second half of the year. This general seasonal timing is confirmed by the posterior concentrations of the in-situ-only inversions. However, yearly reoccurring differences between GOSAT and the in-situ-only based $CO_2$ enhancements from September to November are clearly visible. Thereby, the spread between GOSAT/ACOS and GOSAT/RemoTeC (see also Fig. A1) is much smaller than their difference to and the spread among the in-situ-only inversions. The difference pattern between GOSAT and in-situ-only based $CO_2$ concentrations has already been described by Mengistu and Mengistu Tsidu (2020) and has been shown by Taylor et al. (2022). Furthermore, especially in the second half of the year, different in-situ-only inversions are not consistent as indicated by the large shading in Fig. 2 Panel (a) (see also the individual models in Fig. A2). Reasons for these discrepancies will be further analyzed in Sect. 3.3.

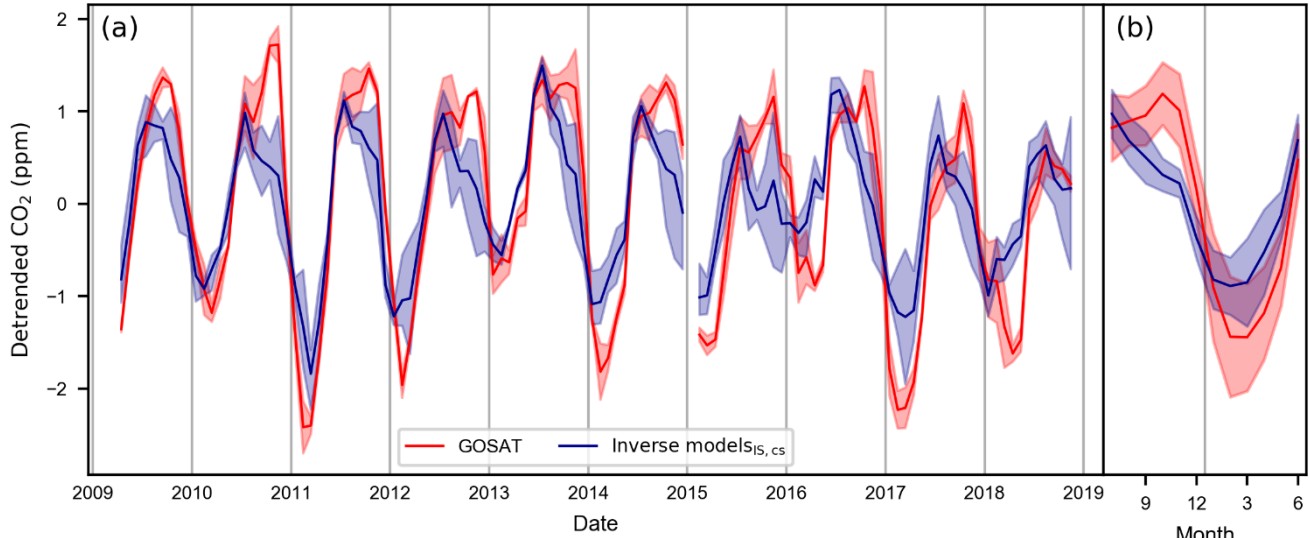

**Figure 2: Monthly southern African detrended $CO_2$ concentrations. GOSAT measured and detrended $CO_2$ concentrations are depicted in red. Modelled posterior $CO_2$ concentrations of three in-situ-only inversions are co-sampled (cs) on GOSAT and depicted as mean in blue. Panel (a) shows the monthly mean $CO_2$ concentrations. The shading indicates the range among the individual ensemble members (GOSAT/ACOS+IS and GOSAT/RemoTeC+IS in red, CT2022, CAMS, and TM5-4DVar/IS in blue). Panel (b) shows the mean seasonal cycle 2009–2018 with the standard deviation over the years as shading.**

For comparison, we additionally use the OCO-2 satellite, which was launched in 2014, and one year of COCCON $CO_2$ column measurements in Namibia. Both datasets show a similar seasonal cycle as seen by GOSAT, i.e. they show concentration maxima later in the year than the in-situ-only inversions (see Fig. A3 and Fig. A4). No other total column

measurement sites (e.g. of the COCCON network or Total Carbon Column Observing Network (TCCON, Wunch et al. (2011)) with coinciding consecutive measurements for more than one year exist in southern hemisphere continental Africa, limiting the validation possibilities of satellite total column measurements in this region.

## 3.2 Southern African top-down and bottom-up CO₂ fluxes

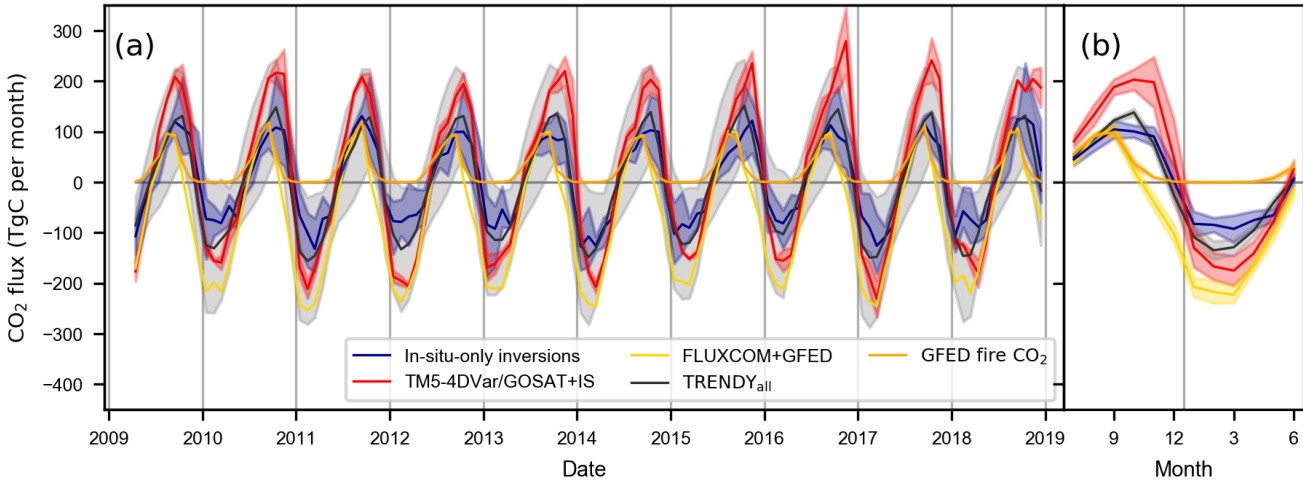

Figure 3: Top-down and bottom-up southern African net CO₂ fluxes. Panel (a) shows the mean monthly net CO₂ fluxes for the southern African region, Panel (b) shows the mean seasonal cycle of the fluxes over the 2009 to 2018 period. The TM5-4DVar/GOSAT+IS fluxes are given in red, in-situ-only inversion fluxes are shown in blue. The mean over all TRENDY models is given in grey. GFED fire emissions are shown in orange and in combination with FLUXCOM NEE in yellow. The shading indicates the range over the GOSAT based fluxes (TM5-4DVar/ACOS+IS and TM5-4DVar/RemoTeC+IS) and the in-situ-only inversion fluxes (CT2022, CAMS, and TM5-4DVar/IS) and the standard deviation over the TRENDY ensemble in Panel (a). In Panel (b) shading indicates the standard deviation over the years. Positive fluxes indicate emissions into the atmosphere. Negative fluxes correspond to an uptake of CO₂ into the land surface.

Assimilating the GOSAT CO₂ concentration measurements in TM5-4DVar, we obtain GOSAT based top-down fluxes in monthly resolution for the study region (see Sec. 2.3.1). As for the concentrations, a clear seasonal cycle is visible (Fig. 3). From January to May CO₂ is taken up by the land surface with a maximum uptake around March. From June to December, CO₂ is released into the atmosphere and reaches a maximum flux in September to November. The number of GOSAT measurements (see Fig. A5 and Fig. A6) is variable throughout the year with the smallest numbers occurring during the rainy season around December and January. This leads to larger uncertainties in the monthly mean satellite CO₂ concentrations and satellite-based fluxes during the transition from maximum to minimum concentrations and fluxes.

A similar timing of the seasonal cycle is also captured by the in-situ-only inversion fluxes (CAMS, CT2022, and TM5-4DVar/IS). However, the in-situ-only inversions' seasonal amplitude is smaller than for TM5-4DVar/GOSAT+IS. To analyze the found differences between TM5-4DVar/GOSAT+IS and the in-situ-only atmospheric inversions, we evaluate the information content provided by the measurements about the southern African carbon fluxes. To this end, we compare the TM5-4DVar fluxes (TM5-4DVar/IS and TM5-4DVar/GOSAT+IS) to the prior fluxes of the inversion model. From Fig. 4 it becomes clear that the in-situ-only fluxes (TM5-4DVar/IS) mainly follow the dynamics of the prior fluxes, whereas the GOSAT based fluxes deviate significantly from the prior. This is expected as the sparse coverage of in situ measurements in Africa and the southern hemisphere in general provides only little information about the African carbon fluxes. In contrast,

satellites provide nearly global coverage of $CO_2$ measurements. Using these measurements in TM5-4DVar, new information about the southern African carbon fluxes can be obtained and lead to a deviation of TM5-4DVar/GOSAT+IS from the prior. This finding also explains the differences among the three in-situ-only inversions (see shaded range of the in-situ-only inversions in Fig. 3). The inversions assume different prior fluxes, which they follow closely, as the information of the in situ data does not substantially inform the inversion.

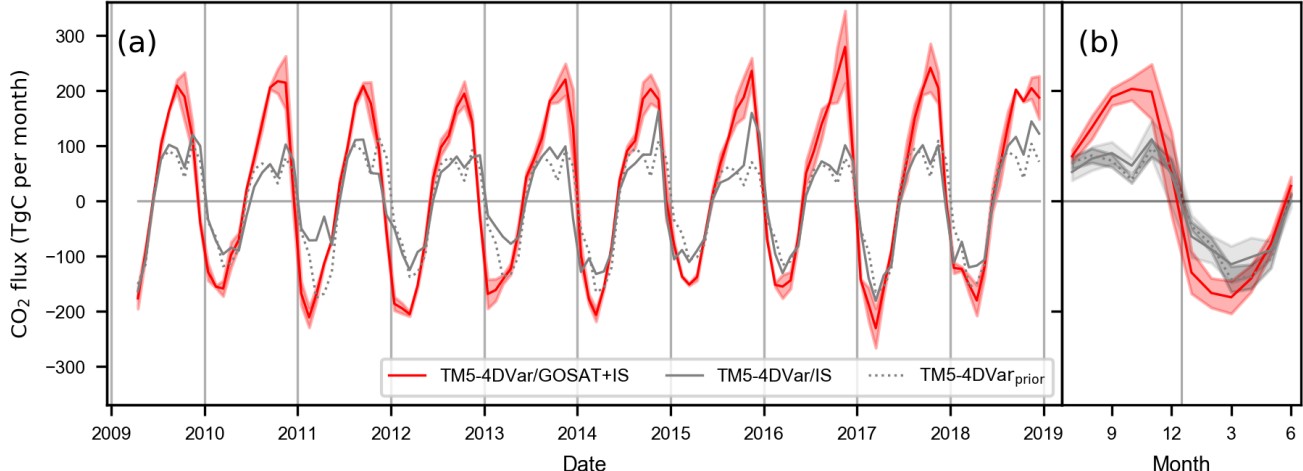

**Figure 4: Top-down southern African net $CO_2$ fluxes from TM5-4DVar. In Panel (a), mean monthly net $CO_2$ fluxes for the southern African region from the TM5-4DVar prior (grey dotted), the in-situ-only inversion TM5-4DVar/IS (grey solid) and the TM5-4DVar/GOSAT+IS inversion (red) are given. Red shading indicates the range of the TM5-4DVar/ACOS+IS and TM5-4DVar/RemoTeC+IS inversions. Panel (b) shows the mean seasonal cycle 2009–2018 with the standard deviation over the years as shading.**

When assimilating OCO-2 satellite measurements instead of GOSAT measurements, the MIP/OCO-2+IS ensemble mean also shows a larger amplitude of the southern African carbon fluxes compared to in-situ-only inversions and MIP/IS (Fig. 5). However, the spread among the MIP/OCO-2+IS models is large, especially during the maximum emissions from September to November. Some models show lower emissions similar to the in-situ-only inversions, whereas others agree with TM5-4DVar/GOSAT+IS. By analyzing the performance of the individual models in these three months, we find that three MIP/OCO-2+IS models reproduce the OCO-2 measurements the best (see Fig. A7) indicating that the OCO-2 measurements were given a considerable weight in the inversion and thus, that the optimized fluxes were informed by measurements (see Text A1). At the same time, these three inversion models (Baker, CAMS, and TM5-4DVar/OCO-2+IS) show the largest $CO_2$ emissions and agree best with TM5-4DVar/GOSAT+IS (see Fig.5 and Fig. A7-A9). Still, their estimated emissions are slightly lower than those of TM5-4DVar/GOSAT+IS. When directly comparing the two TM5-4DVar inversions TM5-4DVar/GOSAT+IS and TM5-4DVar/OCO-2+IS (Fig. 5), the latter has smaller emissions. This is most likely a result of the slightly smaller seasonal amplitude of the $CO_2$ concentrations measured by OCO-2 compared to GOSAT (see Fig. A3).

Concluding, we find that satellite-based inversions, which are actually compatible to the satellite measurements, show larger carbon fluxes in southern Africa than in-situ-only inversions, which suffer from the limited information provided by the

sparse in situ measurements for southern Africa. Our results support current studies (e.g. Basu et al., 2013; Sellers et al., 2018; He et al., 2023) reporting that satellite observations do well inform atmospheric inversions for flux estimates on sub-continental scales. Satellite $CO_2$ concentration measurements, therefore, provide a unique information source and are especially valuable in regions with sparse in situ measurement coverage. The already long record provided by GOSAT will be more and more complemented over time by the growing record of OCO-2 and future $CO_2$ sensors providing even more

extensive measurements.

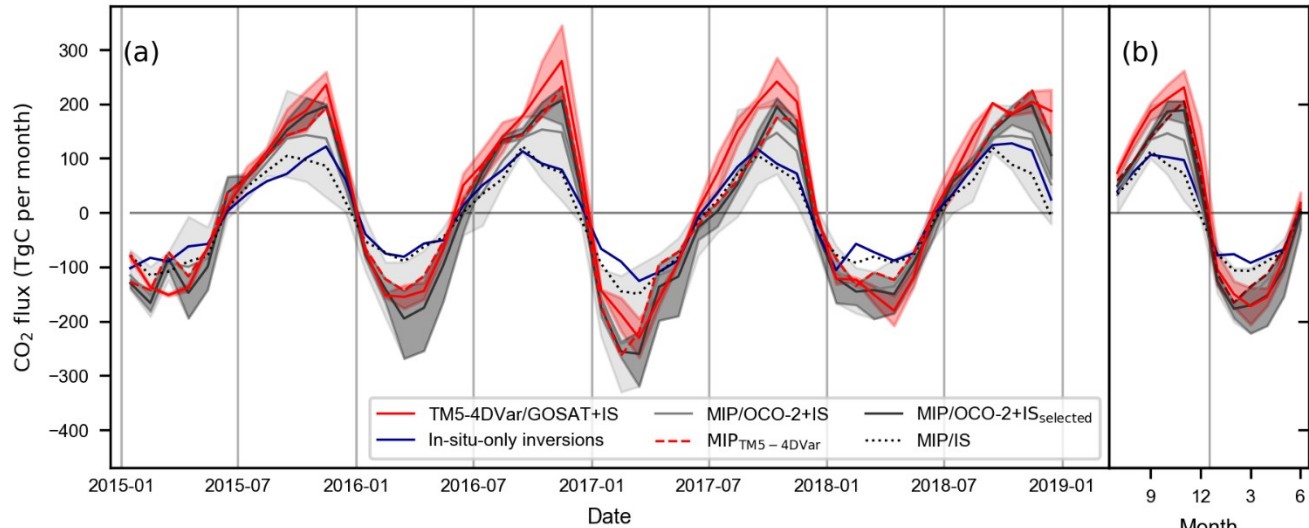

**Figure 5: Top-down southern African net $CO_2$ fluxes from MIP.** In Panel (a) mean monthly net $CO_2$ fluxes for the study region are given by TM5-4DVar/GOSAT+IS in red, the MIP/OCO-2+IS ensemble mean in grey, the mean over three selected MIP models (CAMS, TM5-4DVar, and Baker) in black, and TM5-4DVar/OCO-2+IS as part of the MIP ensemble in red dashed. In-situ-only
inversion fluxes are given in blue as mean of CAMS, CT2022 and TM5-4Dvar/IS and in black dotted from the MIP/IS ensemble. The shading indicates the range over the GOSAT fluxes (TM5-4DVar/ACOS+IS and TM5-4DVar/RemoTeC+IS), the MIP ensemble, and the three selected MIP models. Panel (b) gives the mean seasonal cycle from 2015 to 2018 with shading indicating the range over the MIP ensembles' models and the standard deviation of the TM5-4DVar/GOSAT+IS over the years.

Next to the in-situ-only inversion fluxes, we compare the TM5-4DVar/GOSAT+IS fluxes to FLUXCOM $CO_2$ fluxes. As
FLUXCOM only provides NEE fluxes, we add GFED fire $CO_2$ emissions to obtain an NBP estimate. In Fig. 3, FLUXCOM+GFED only reaches positive monthly fluxes from June to September due to fire emissions occurring during that time. From October to May it shows a net $CO_2$ uptake. While the timing of the maximum sink agrees well between FLUXCOM+GFED and the inversion fluxes, FLUXCOM+GFED shows a smaller amplitude and an earlier drop in emissions compared to TM5-4DVar/GOSAT+IS and in-situ-only inversion fluxes. The tendency of FLUXCOM to report a
stronger carbon sink on the southern hemisphere compared to other datasets is described in Jung et al. (2020). It is expected that the sparsity of eddy-covariance towers in Africa or in similar ecosystems hampers the machine-learning based approach

of FLUXCOM for estimating $CO_2$ fluxes in the study area. Jung et al. (2020) describe larger uncertainties due to representation errors in semi-arid regions.

Finally, we compare the inversion results to the ensemble of process-based vegetation models of the TRENDYv9 project.
The mean of the DGVM ensemble in Panel (a) of Fig. 3 shows a smaller amplitude than the GOSAT fluxes and compares with the in-situ-only inversion fluxes. However, as indicated by the large standard deviation, the models deviate substantially from each other. Foster et al. (2024) and Metz et al. (2023) observed a similar large spread among DGVMs for the North American Temperate region and Australia, respectively. Both studies highlight the importance of performing a sub-selection of DGVMs agreeing well with atmospheric $CO_2$ measurements.

### 3.3 GOSAT and SIF atmospheric constraints on TRENDY models

Given the large spread of the TRENDY models, we select DGVMs according to their agreement with the GOSAT based $CO_2$ fluxes and SIF. Thereby, in a first step, we compare the monthly mean DGVM and TM5-4DVar/GOSAT+IS NBP and NEE fluxes based on the RMSE of the monthly fluxes and the agreement in the seasonality. In a second step, only for the well matching DGVMs, we additionally compare the GPP normalized mean seasonal cycle to the GOME SIF normalized mean seasonal cycle. Only models with a timing of the minimum and maximum GPP agreeing within +-1 month with the normalized SIF seasonal cycle are selected (see Fig. 6). This ensures the correct seasonal timing of the modelled GPP fluxes. Based on these criteria, we select the models ORCHIDEE (RMSE NBP: 60.2 TgC/month, RMSE NEE: 68,2 TgC/month), ORCHIDEEv3 (RMSE NBP 70.2 TgC/month, RMSE NEE: 56.2 TgC/month) and CABLE-POP (RMSE NBP: 78.2 TgC/month, RMSE NEE: 63.6 TgC/month). All other models, except for the model OCN, already were excluded in the first step of NBP/NEE comparison. OCN performs well in the NBP/NEE comparison but shows larger deviations in the SIF/GPP comparison (see Fig. 6). Therefore, it was excluded in the second selection step and is not included in the TRENDY selection. The exclusion of OCN underlines the importance of the SIF/GPP selection and demonstrates that a correct timing of the net $CO_2$ exchange fluxes does not necessarily imply the correctness of the modelled gross fluxes. In general, it is noteworthy that only three out of 18 TRENDY models pass our selection process. This again reveals the large uncertainties associated with the TRENDY ensemble estimate for semi-arid southern hemispheric Africa.

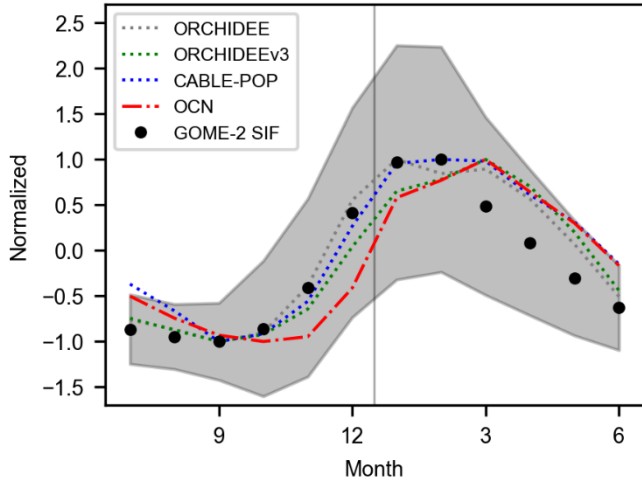

**Figure 6: Seasonal cycle of SIF and selected TRENDY models. The normalized mean seasonal cycle of GOME-2 SIF (2009–01/2018), the three selected DGVMs (ORCHIDEE, ORCHIDEEv3, CABLE-POP) GPP, and OCN GPP (2009–2018) are shown in solid black, colored dotted, and red dashdotted, respectively. The spatial standard deviation over monthly GOME-2 SIF aggregated on a 1°x1° is given as shading.**

The NBP mean over these three models is given in Panels (a) and (b) of Fig. 7. The models reproduce the timing and strength of the TM5-4DVar/GOSAT+IS NBP fluxes. Only at the beginning of the emission period around July to September, the TRENDY selection fluxes are lower. Furthermore, the selection shows a significantly smaller sink in 2012 and smaller source in 2016. Note that ORCHIDEE is part of the TRENDY selection and is also used by the in-situ-only inversion CAMS as prior flux assumption. This explains why CAMS best matches TM5-4DVar/GOSAT+IS $CO_2$ fluxes and GOSAT $CO_2$ concentrations (see Fig. A2 and Fig. A7, respectively).

Fire emissions contribute substantially to the seasonality of the southern African carbon fluxes. They largely explain the beginning of the emission period from July to September (see Fig. 3). Different fire emission data products differ significantly and suggest large uncertainties on the magnitude of the actual fire emissions in our study region (see Fig. A10). GFED, which we use for our analyses, shows the largest fire emissions but could even underestimate the actual emissions as suggested by current literature for southern hemispheric Africa (Ramo et al., 2021, van der Velde et al., 2024).

To exclude the influence of fire emission in the comparison, we analyze the monthly NEE fluxes of the TRENDY selection compared to the TM5-4DVar/GOSAT+IS NBP fluxes with GFED fire emissions subtracted. The subtraction of the fire emissions leads to a better agreement between both datasets, especially at the beginning of the emission period suggesting that fire fluxes in the DGVMs do not agree to the GFED fire fluxes (see Fig. 7 Panels (c) and (d)). This goes along with large uncertainties in DGVM fire fluxes being reported previously (Bastos et al., 2020).

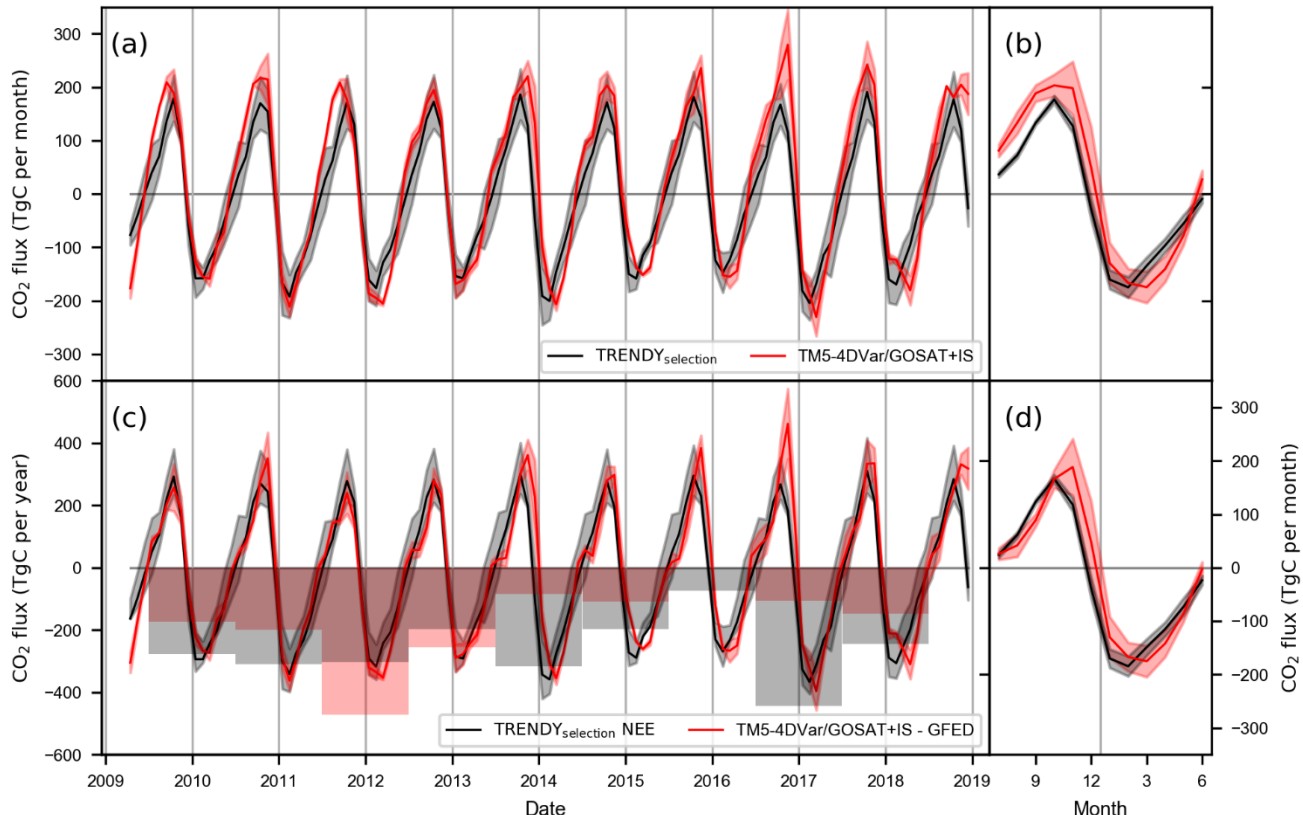

**Figure 7: Annual and mean monthly NBP and NEE fluxes in southern Africa. The NBP fluxes by TM5-4DVar/GOSAT+IS (red) and selected TRENDY models (black) are given as mean monthly fluxes in Panel (a) and as the mean seasonal cycle in Panel (b). Similar to that, Panel (c) and (d) show the monthly NEE fluxes (GFED is subtracted from TM5-4DVar/GOSAT+IS). Additionally, the annual (July to June) NEE fluxes of the selected TRENDY models and TM5-4DVar/GOSAT+IS – GFED fluxes are given. The shading indicates the standard deviation over the TRENDY models and range of TM5-4DVar/ACOS+IS and TM5-4DVar/RemoTeC+IS (Panel (a) and (c)) and over the years (Panel (b) and (d)).**

Panel (c) in Fig. 7 additionally shows the annual NEE fluxes (July-June) as bars. The absolute difference between TM5-4DVar/GOSAT+IS and TRENDY annual fluxes is large in some years. These differences are caused by a stronger sink at the beginning of 2012 and enhanced emissions at the end of 2013 and 2016 in TM5-4DVar/GOSAT+IS compared to TRENDY. However, while both datasets do not agree on the absolute value of annual fluxes in most of the years, they show a similar IAV. Both datasets show a slightly stronger $CO_2$ uptake from 2010 to 2012. These years were strong and moderate La Niña years with enhanced rainfall in 2010 and 2011 in the study region compared to the longtime mean (see Fig. A11). Additionally, lower than average temperatures led to enhanced soil moisture near the surface in 2010/11. The soil moisture declined in 2012 to reach the long-term average. In 2015 and 2016, the sink given by the GOSAT and TRENDY selection NEE fluxes is small. These two years have been a weak and a strong El Niño year respectively with dry conditions and in case of 2016 exceptionally high temperatures (see Fig. A11). These findings agree well with results from Pan et al. (2020)

pointing out that temperature and precipitation extremes impact the African ecosystems heavily and therefore play a key role in the African carbon fluxes.

To conclude, especially the monthly NEE and NBP fluxes, but also the IAV of the selected TRENDY models agree well with TM5-4DVar/GOSAT+IS NEE and NBP – although the latter was not a criterion in the selection process of the TRENDY models. This suggests that the selected models indeed capture the carbon cycle dynamics even on a decadal time

scale. For this reason, we use the model selection for further investigations of vegetation processes driving the southern African carbon cycle.

## 3.4 Seasonal and IAV of TRENDY gross fluxes

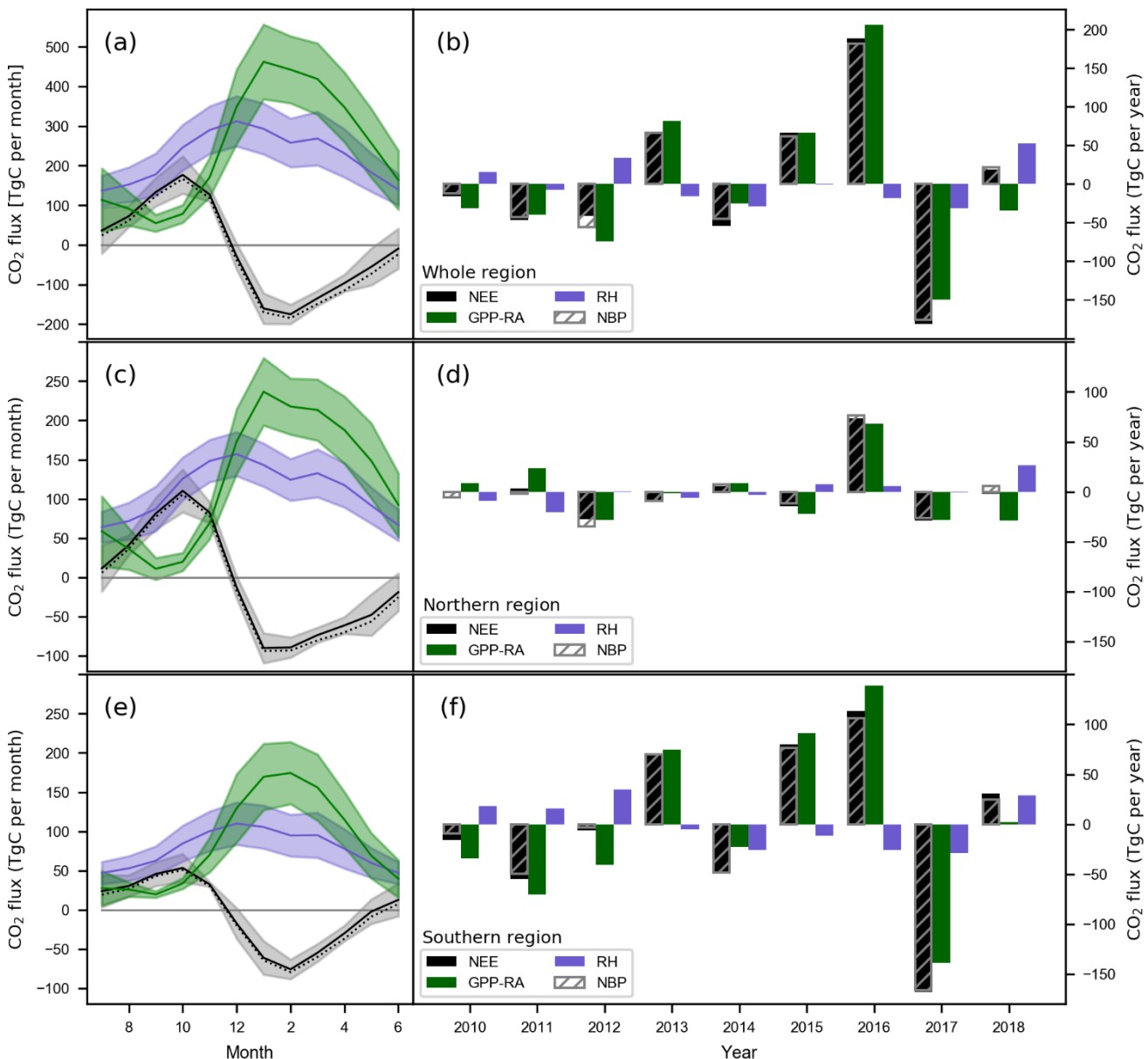

**Figure 8: Annual and mean monthly CO₂ net and gross fluxes. The mean monthly fluxes (Panel (a), (c), (e)) and annual (July to June) anomalies (Panel (b), (d), (f)) of NBP, NEE, GPP-RA, and RH of the selected TRENDY models are given in black, grey (dotted), green, and blue, respectively. The fluxes are given for the whole study region (Panel (a) and (b)), the Savanna dominated northern region (north of 17°S, Panel (c) and (d)), and the southern region with grass- and shrubland (Panel (e) and (f)). The annual anomalies are calculated by subtracting the individual long-term mean of the annual fluxes. Thereby, a positive GPP anomaly denotes a reduced GPP and vice versa. The shading in Panel (a), (c), and (e) indicates the standard deviation over the three selected models (ORCHIDEE, ORCHIDEEv3, and CABLE-POP).**

To investigate the vegetation dynamics shaping the seasonal cycle of the southern African $CO_2$ exchange, we use the selected TRENDY models to further split up the net ecosystem exchange fluxes into the gross fluxes NPP (GPP – RA) and RH. The gross and net fluxes are given as mean seasonal cycle and annual anomalies in Fig. 8. In the mean seasonal cycle for the whole study region (Panel (a)), we can see a clear difference in timing between RH and GPP-RA. Heterotrophic respiration increases early in September and October, while RA increases one to two months later simultaneously with GPP (see Fig. A12). The dephasing between RH and GPP-RA leads to a prolonged emission phase in the net $CO_2$ exchange. It takes place in the whole region and occurs in the savanna dominated north (Fig. 8 Panel (c)) and in the grass- and shrublands in the south (Panel (e)). The dephasing takes place in every year (see Fig. A13) and is present in all selected TRENDY models. It causes a mean $CO_2$ release of 494 TgC during the emission phase, which is about 17% and 18% of the annual total RH and GPP-RA, respectively. When looking at the monthly precipitation over the study region (see Fig. A14) one can identify a distinct drought phase occurring in the whole study region. The subsequent start of the rainy season in September and October temporally coincides with the early increase in RH. This finding resembles the results of Metz et al. (2023) in Australia, describing an increase of soil respiration with the beginning of the rainy season prior to the start of the growing season. The study finds soil respiration pulses resulting from rewetting of soils to cause the continental scale increase of soil respiration. Such soil respiration pulses at local arid sites are discussed in the context of the Birch effect (Birch et al., 1964; Jarvis et al., 2007). Thereby the rewetting of the soil enables microbial populations to grow and to transform the carbon stored in the soils into $CO_2$ emissions. $CO_2$ is then released in substantial amounts within a short period of time. Like in Metz et al. (2023), we find short duration emission pulses in the daily flux record of a FLUXNET station in the study region. Exemplary annual records of the FLUXNET station in the Kruger National Park (Archibald et al., 2009) show $CO_2$ emission caused by precipitation pulses (see Fig. A15). This is also reported in Fan et al. (2015) studying a two-year measurement record of carbon fluxes in Kruger National Park in more detail. The study finds recurring respiration emission pulses due to precipitation events and attributes them to the Birch effect. The TM5-4DVar/GOSAT+IS fluxes indicate an even larger time lag between the increase of soil respiration and NPP in some years compared to TRENDY. A prolonged emission phase of an additional 1 to 2 months (see Figure 7, Panel (c)) takes place in years with especially low soil moisture (2013, 2015, 2016, see Fig. A11). This later drop in emissions could either be caused by a delayed start of the GPP rise in the growing season or enhanced soil respiration due to the drier conditions causing an enhanced accumulation of soil carbon during the years. It is not possible to investigate this further, as none of the TRENDY DGVMs captured the IAV in the timing of the emission phase.

It is noteworthy that large parts of the not selected 'other' TRENDY models miss the dephasing between RH and GPP-RA. Their NBP estimates, therefore, do not agree with the emissions around October found by the satellite inversion. Implementing soil respiration due to rewetting more accurately in those models could improve their agreement with the satellite-based fluxes. Metz et al. (2023) found that the dephasing in the TRENDY models is most likely caused by a different response time of soil respiration and vegetation growth on precipitation e.g. water needs to percolate into the deeper soil layers with plant roots to initiate plant growth, whereas heterotrophic respiration is driven by upper soil layer soil

moisture or precipitation. The implementation of such a time lag between heterotrophic respiration and GPP seems to be a necessary but not sufficient prerequisite to accurately capture the seasonal carbon flux variability in semi-arid southern Africa. Our results call for studies on how to implement the response of ecosystems on soil rewetting more accurately to improve the consistency and accuracy of the TRENDY ensemble in semi-arid regions.

Looking at the annual gross flux anomalies given by the TRENDY selection (Fig. 8, Panel (b)), we see that the IAV of NBP and NEE is driven by GPP mainly. Enhanced GPP from 2010 to 2012 leads to a constant stronger uptake of $CO_2$. In 2017 a strongly enhanced GPP causes a large $CO_2$ sink. Reduced GPP in 2013, 2015, and 2016 results in positive NEE anomalies associated with a reduced sink in NEE. RH only plays a minor role and mostly slightly counteracts the GPP anomalies. These findings agree with the studies of Ciais et al. (2009), Weber et al. (2009), and Williams et al. (2008) which identify GPP variability as a major source of African fluxes' IAV. It is, however, in contrast to semi-arid Australia, where Metz et al. (2023) found large IAV of RH driven by precipitation anomalies during the dry season. The African study region, however, has a distinct and regular dry season every year (see Fig. A14), leading to a smaller influence of RH on IAV. Note that in 2017, GOSAT suggests a much smaller annual $CO_2$ sink. However, the discrepancy is mainly caused by a significant difference in the emissions in the second half of the year and while both datasets agree well in the phase of carbon uptake (see Fig. 7, Panel (c)). Therefore, the TM5-4DVar/GOSAT+IS fluxes support the large GPP anomaly given by the TRENDY models but suggest stronger respiration or fire fluxes at the end of 2016.

Looking at the subregions (Panels (d) and (f)), one can see that the sinks in 2010, 2011 and 2017 are mainly driven by the southern grassland region, where enhanced precipitation occurred during these years (see Fig. A11). The comparably large release in 2016 seems to be driven by the whole African region experiencing the highest annual temperatures and driest conditions within the 10-year study period. Therefore, GPP IAV seems to be heavily impacted by precipitation variability. According to GFED (see Fig. A10), fire emissions play a minor role in impacting GPP and driving NBP anomalies. The variability of fire emissions is much lower than for NBP and GPP-RA. In the whole study region, IAV (calculated as standard deviation over the years) of GPP-RA and NBP fluxes are 97.7 TgC/year and 94.1 TgC/year, respectively. IAV of GFED fire emissions is 27.3 TgC/year, similarly low as IAV of RH (27.1 TgC/year). Furthermore, the annual fire emissions do not amplify the trend of the NBP anomalies. They have been on a normal level during the large positive NBP anomaly in 2016. Higher than average fire emissions counteract the sink anomalies in 2011 – 2012 and only the slightly reduced fires in 2017 amplify the sink anomaly.

**4 Conclusions**

The sparsity of in situ $CO_2$ concentration and flux measurements cause large uncertainties in carbon flux estimates in the southern African region. We show that satellite measurements provide additional information leading to an improvement of our knowledge about the southern African carbon cycle. Our study demonstrates that satellite measurement based

atmospheric inversions and SIF can be used as atmospheric constraints for sub-selecting TRENDY DGVMs. This is necessary as TRENDY flux estimates show a large spread in our study region.

Using the satellite based selection of TRENDY DGVMs, we find that IAV of NBP and NEE in southern Africa is driven by GPP variability. This supports findings by Ciais et al. (2009), Weber et al. (2009), and Williams et al. (2008) using individual vegetation models. The enhancements in annual GPP mainly originate in the grass- and shrublands in the southern part of the study region and occur in years with enhanced amount of precipitation. The seasonal variability of the southern African carbon fluxes is impacted by soil respiration dynamics, which are driven by the onset of the rainy season. Respiration pulses have been reported under the term of the Birch effect for arid Africa (Fan et al., 2015) and have been shown to be relevant on continental scale in semi-arid Australia (Metz et al., 2023). This enforces the relevance of rain-induced $CO_2$ emissions for the southern African region and semi-arid regions in general. Our results emphasize the importance of correctly representing the response of semi-arid ecosystems to soil rewetting in DGVMs (e.g. different response times of RH and GPP), as this was found to be a prerequisite to accurately capture the seasonal carbon cycle dynamics.

**Appendix A**

Text A1: The performance of the individual MIP models.

In Fig. 5, the ensemble mean of MIP/OCO-2+IS shows lower emissions than TM5-4DVar/GOSAT+IS in the second half of the year. A selection of three models (Baker, TM5-4DVar, and CAMS), however, shows larger fluxes and agrees better with the GOSAT based fluxes (see Section 3.2 and Fig. 5). Next to the OCO-2 informed posterior fluxes used for the analysis in the main text, the MIP/OCO-2+IS dataset provides the prior fluxes used by the individual MIP models. Furthermore, 5% of the OCO-2 measurements are withheld for validation purposes and modelled $XCO_2$ co-sampled on the left-out measurements are provided for each model but CSU. The OCO-2 co-samples and the prior fluxes of the MIP models can be used to further evaluate the differences between the three selected models and the other MIP models.

In Fig. A7, the mismatch between $XCO_2$ modelled by MIP and $XCO_2$ measured by OCO-2 is given for the months of the strongest emissions (September to November). The $XCO_2$ mismatch is the smallest for the three selected models, Baker, TM5-4DVar, and CAMS, which have at the same time the smallest mismatch to TM5-4DVar/GOSAT+IS. Hence, the models which reproduce the OCO-2 measurements best, also agree best with the GOSAT based $CO_2$ fluxes.

The difference between posterior and prior fluxes for the MIP models are given in Fig. A8. TM5-4DVar and Baker have the largest differences between the posterior and prior fluxes. Therefore, it is likely that even though the prior fluxes of TM5-4DVar and Baker deviate strongly from the GOSAT based fluxes (see Fig, A9), considerable weight was given to the OCO-2 measurements in the inversion. As a result, the posterior fluxes are closer to the GOSAT based fluxes than to their prior fluxes (Fig. A8). As the CAMS prior already agrees reasonably well with TM5-4DVar/GOSAT+IS fluxes, no conclusion on the weights can be drawn here.

The other MIP models, which have lower emission fluxes, show larger mismatches to the OCO-2 $XCO_2$ measurements for September to November (Fig. A7). Although, for most of these models, assimilating OCO-2 increases the emission fluxes and reduces the difference to the GOSAT based fluxes (see Fig. A8 and Fig. A9), the changes (i.e. the difference between posterior and prior fluxes) are small compared to TM5-4DVar and Baker (see Fig. A8). The larger mismatch to OCO-2 $XCO_2$ and the smaller posterior – prior flux differences seem to indicate that a smaller weight was given to the OCO-2 measurements compared to the selected MIP models.

In general, the GOSAT flux mismatch and the OCO-2 $XCO_2$ mismatch is larger in October and November than in September. This is most likely caused by the prior fluxes in September already being closer to the GOSAT based fluxes than in the other two months (see panel b in Figure A9).

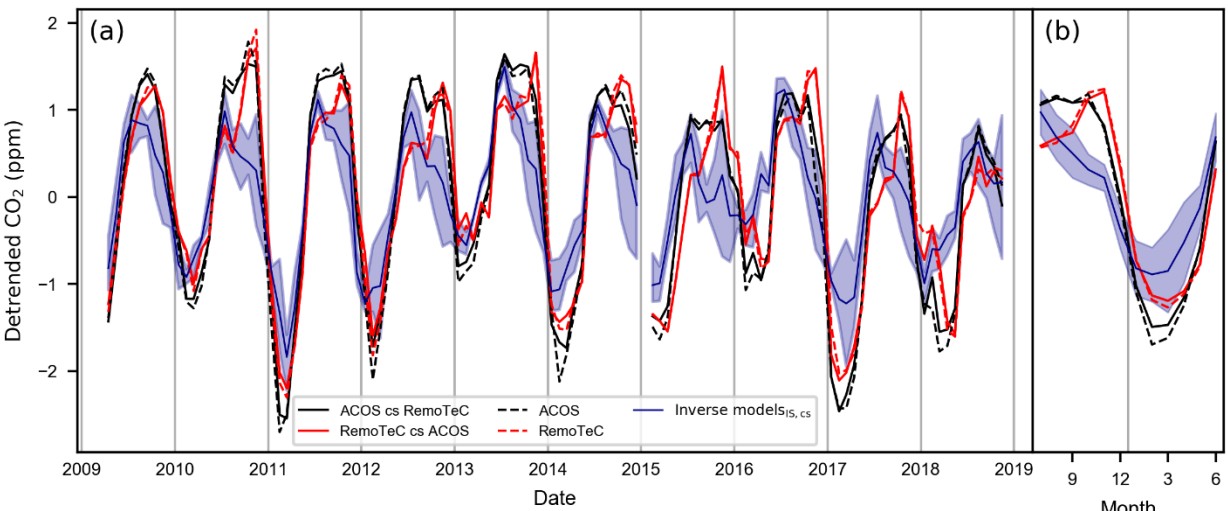

**Figure A1: Monthly southern African detrended CO₂ concentrations measured by GOSAT. GOSAT/ACOS is given in black, GOSAT/RemoTeC is given in red. Dashed lines show the mean CO₂ concentrations over the whole dataset. The mean CO₂ concentrations of the soundings included in both datasets, ACOS and RemoTeC, are given as solid line. CS stands for co-sampled and indicates that only soundings, also included in the other dataset are considered. The deviations due to different sampling are in sub-ppm scale and do not explain the differences between ACOS and RemoTeC. Modelled posterior CO₂ concentrations of the in-situ-only inversions are co-sampled (cs) on GOSAT and depicted as mean in blue for comparison. The shading indicates the range among the individual in-situ-only inversions. Panel (b) shows the mean seasonal cycle 2009–2018 with the standard deviation over the years as shading.**

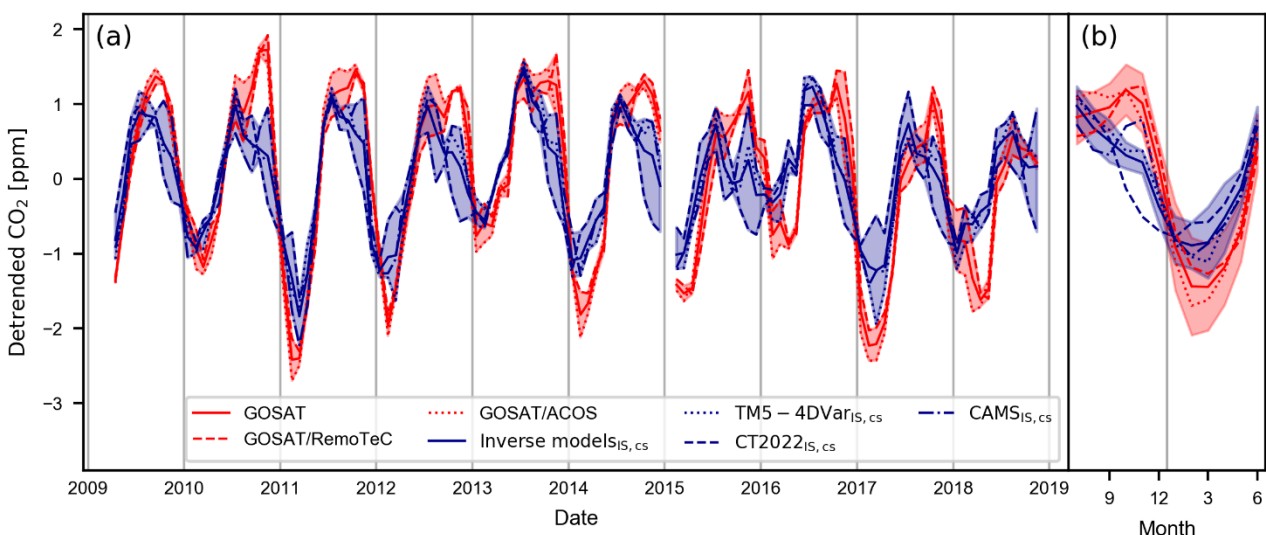

**Figure A2: Monthly southern African detrended CO₂ concentrations given by inversions and satellites. Like Fig. 1, but with detrended XCO₂ of individual in-situ-only inversions co-sampled (cs) on the GOSAT measurements in dark blue (CT2022 dashed, CAMS dash-dotted, and TM5-4DVar/IS dotted). Panel (a) gives the monthly mean CO₂ concentrations, while Panel (b) shows the mean seasonal cycle 2009-2018. The shading indicates the range among GOSAT/ACOS and GOSAT/RemoTeC and the range among the three in-situ-only inversions in Panel (a). In Panel (b) the shading indicates the standard deviation over the year.**

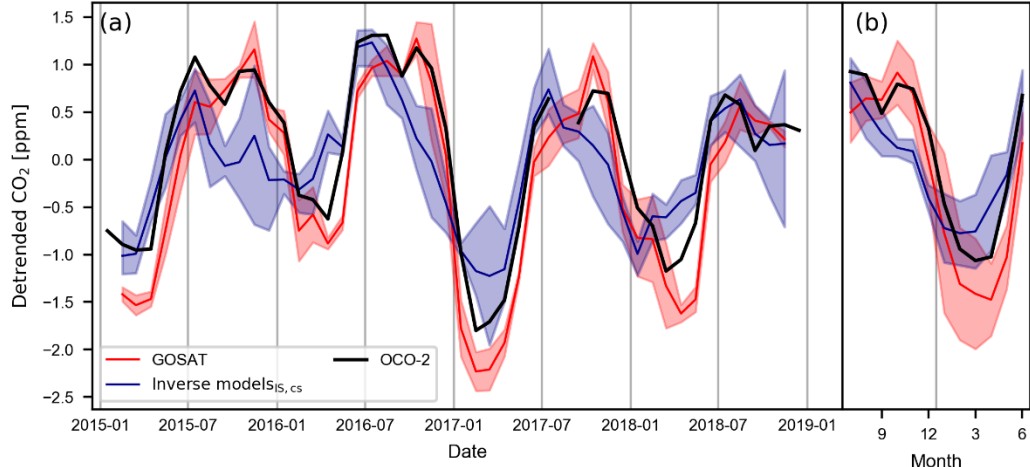

**Figure A3: Monthly southern African detrended CO₂ concentrations given by inversions and satellites. Like Fig. 1, but with detrended XCO₂ measurements of OCO-2 in black for the time period from 2015 to 2018. Panel (a) gives the monthly mean CO₂ concentrations, while Panel (b) shows the mean seasonal cycle 2015-2018. The shading indicates the range among GOSAT/ACOS and GOSAT/RemoTeC and the range among the three in-situ-only inversions in Panel (a). In Panel (b) the shading indicates the**
**standard deviation over the years.**

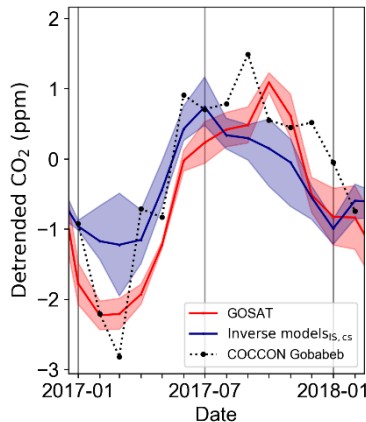

**Figure A4: Monthly southern African detrended CO₂ concentrations given by inversions, satellites and COCCON measurements.**
Like Fig. 1, but only for 01/2017-02/2018 and with detrended XCO₂ measurements of the COCCON stations Gobabeb in black. The full dataset of COCCON measurements is used, without performing a co-sampling on GOSAT measurements nor further filtering.

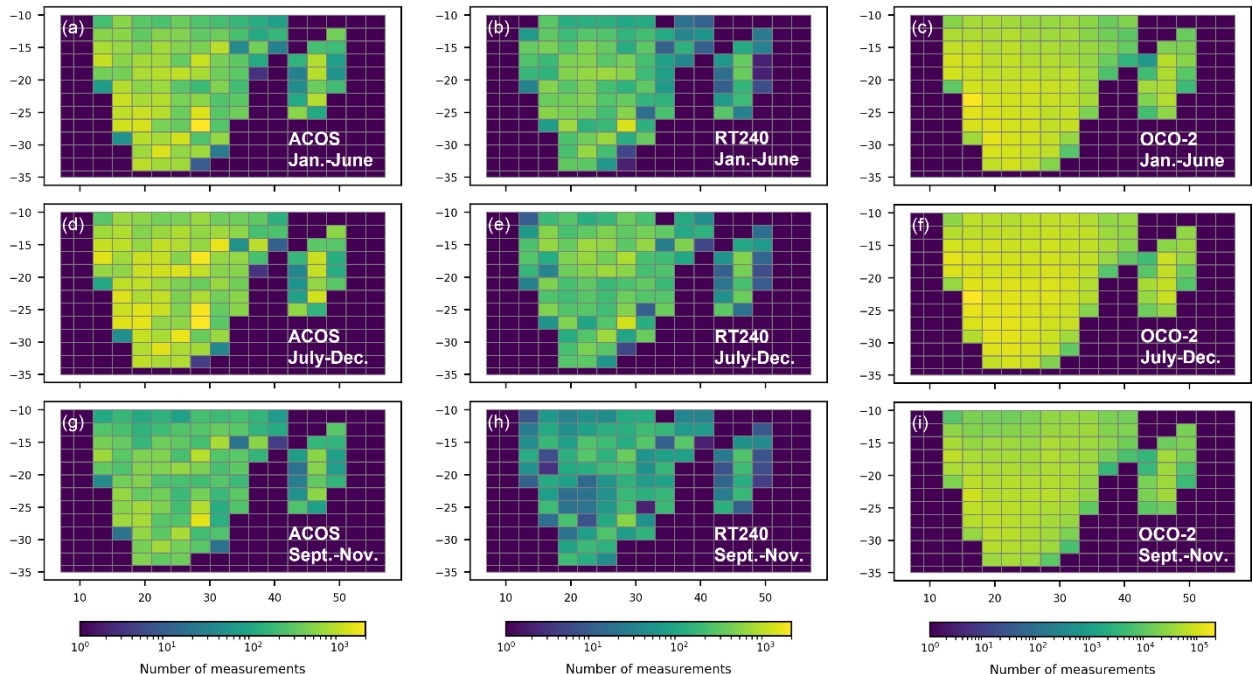

**Figure A5: Number and distribution of satellite CO₂ concentration measurements above southern Africa. (a), (d), and (g) Total number of GOSAT/ACOS, (b), (e), and (h) GOSAT/RemoTeC, and (c), (f), and (i) OCO-2 data per 3°x2° grid cell for (a) - (c) the months of carbon uptake (January – June), (d) - (f) the emission season (July – December), and (g) – (i) the month with the strongest emissions. GOSAT/ACOS and GOSAT/RemoTeC measurements from 2009 to 2018 and OCO-2 measurements from 09/2014 to 2018 are included. The maximum of the color scale is the same for all time periods, but different for OCO-2 than for GOSAT/ACOS and GOSAT/RemoTeC. Compared to GOSAT/ACOS, GOSAT/RemoTeC has a reduced number of measurements, as RemoTeC algorithm applies stricter filtering of the GOSAT soundings.**

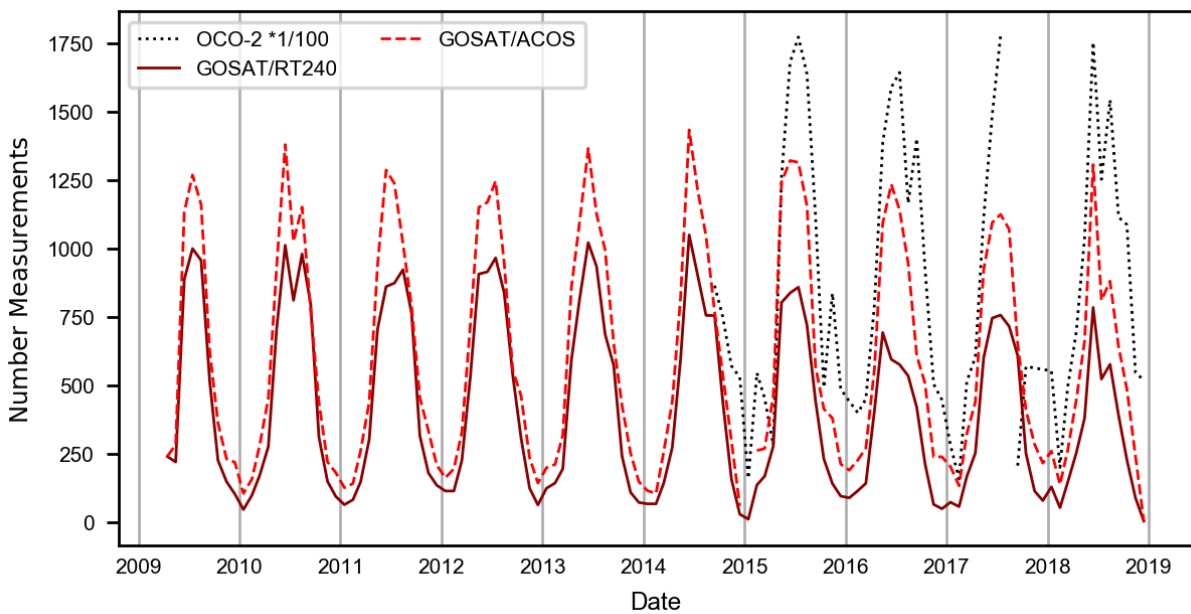

**Figure A6: Number of satellite measurements per month. The amount of satellite measurements of the GOSAT/ACOS (red dashed), GOSAT/RemoTeC (dark red solid), and OCO-2 (grey dotted) dataset are given. Note that the number of OCO-2 measurements is shown divided by 100 to enable a comparison to the much less abundant GOSAT measurements.**

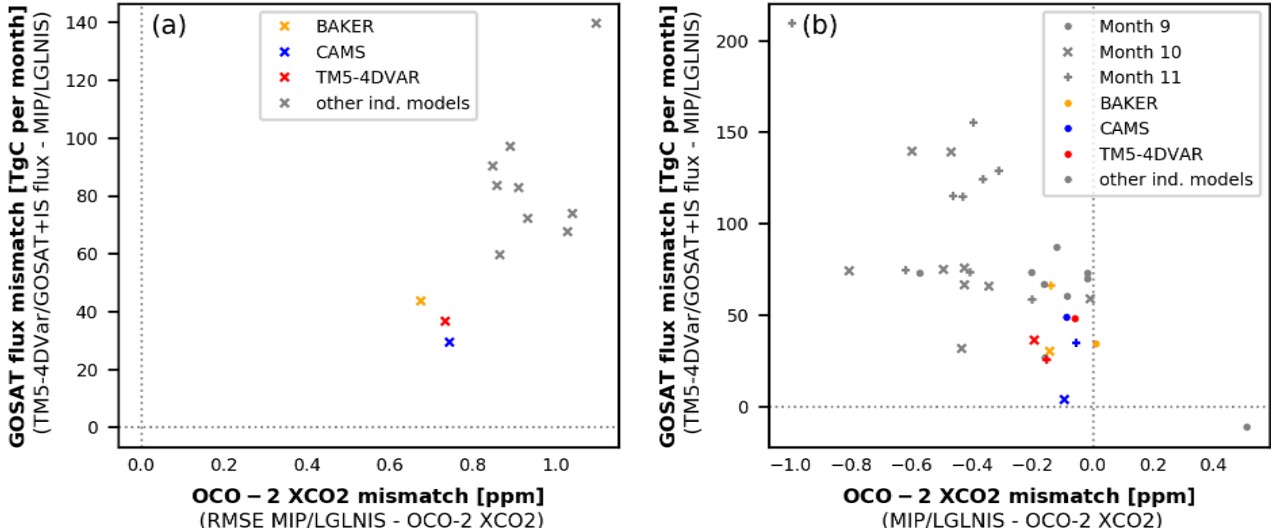

**Figure A7: Mismatch between GOSAT-informed and OCO-2 informed fluxes versus mismatch between OCO-2 informed simulated XCO2 and OCO-2 measured XCO2.** For the MIP/OCO-2+IS inversions 5% of the OCO-2 measurements are withheld for validation purposes and modelled XCO2 co-sampled on the measurements are provided for each model but CSU. Panel (a) gives the RMSE of the OCO-2 measurements and the modelled co-sampled XCO2 from September to November for each model. In panel (b), the mean differences of the OCO-2 measurements and modelled co-samples for each month and model are given. In both panels, the OCO-2 XCO2 mismatch is plotted against the difference of the monthly TM5-4DVar/GOSAT+IS and individual MIP/OCO-2+IS CO2 fluxes for the strongest emission period from September to November. The MIP models Baker, CAMS and TM5-4DVar are highlighted in yellow, blue and red. The other individual MIP models are given in grey. The three highlighted models show the smallest OCO-2 XCO2 mismatch and the smallest difference to the monthly fluxes of TM5-4DVar/GOSAT+IS (exception Baker in September, panel b).

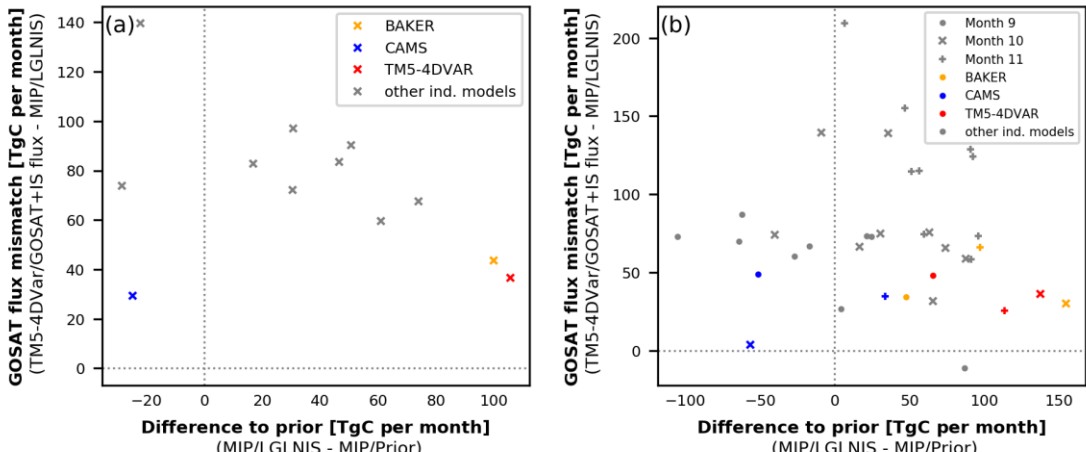

**Figure A8: Mismatch between GOSAT-informed and OCO-2 informed fluxes versus difference between OCO-2 informed fluxes and model prior fluxes. The individual MIP models differ in their assumed prior fluxes. In this figure, the differences of the monthly posterior to the prior fluxes (x-axis) and to the GOSAT based fluxes (TM5-4DVar/GOSAT+IS, y-axis) are compared. Differences are calculated using the monthly flux over the whole study region and the time period 2015-2018. Panel (a) shows the mean over September to November, the time of the strongest $CO_2$ emissions. In panel (b), the differences are given for each of the three individual months. The MIP models Baker, CAMS, and TM5-4DVar are highlighted in yellow, blue, and red. The other individual MIP models are given in grey.**
**For most of the models the assimilation of OCO-2 measurements increases the mean monthly fluxes from September to November (difference to prior larger than zero). Only for CAMS and UT, and for some models in September, the mean posterior fluxes are smaller than the prior fluxes.**

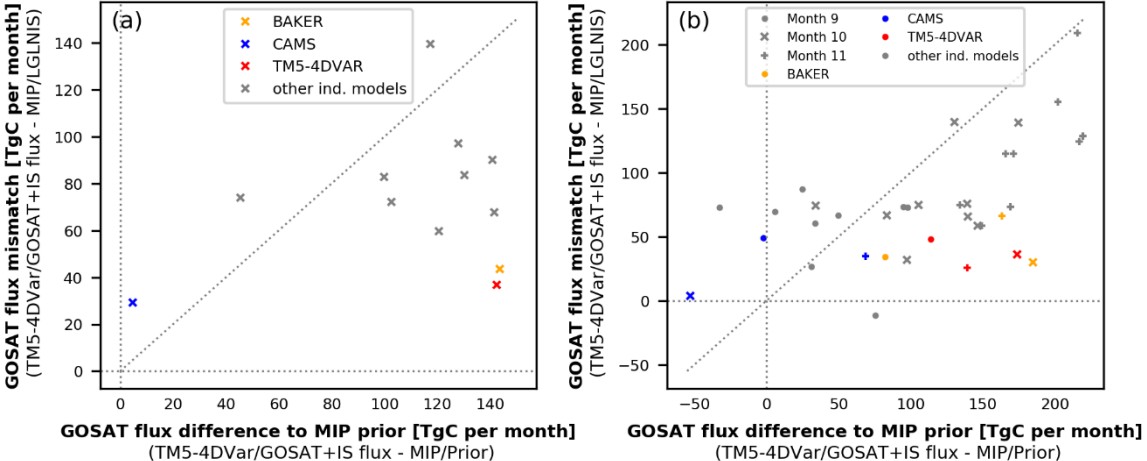

**Figure A9: Mismatch between GOSAT-informed and OCO-2 informed fluxes versus difference between GOSAT informed fluxes and OCO-2 MIP prior fluxes.** The differences of the monthly GOSAT inversion fluxes (TM5-4DVar/GOSAT+IS) to the MIP posterior (y-axis) and MIP prior fluxes (x-axis) for the individual MIP models is given. Panel (a) gives the mean differences for the months September to November. Panel (b) shows the differences for the individual months. The MIP models Baker, CAMS and TM5-4DVar are highlighted in yellow, blue and red. The other individual MIP models are given in grey. The 1:1 line is given in grey dotted. For most of the MIP models, assimilating OCO-2 reduces the flux difference to the GOSAT based fluxes (i.e. markers are below the 1:1 line).

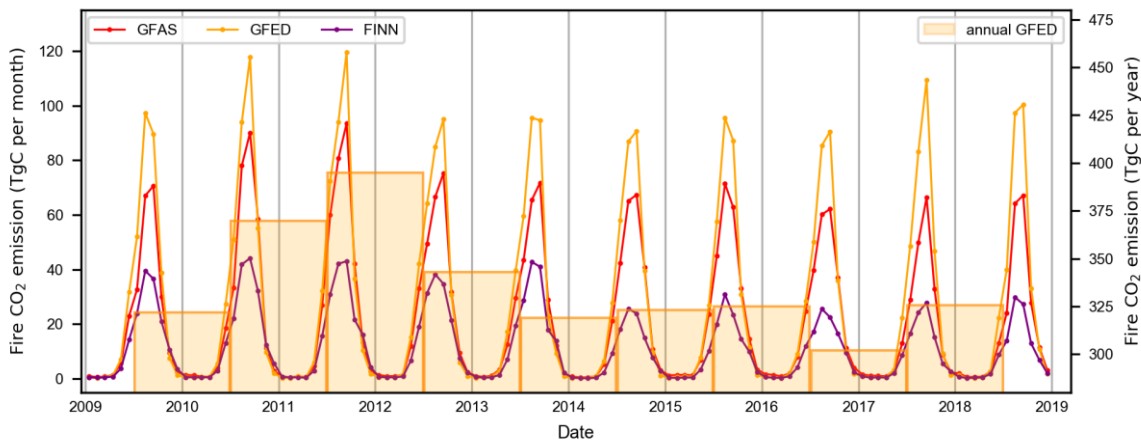

**Figure A10: CO₂ fire emissions in southern Africa. The monthly CO₂ fire emissions collected by three fire emission databases (GFED in orange, Global Fire Assimilation System (GFAS, Kaiser et al., 2012) in red and the Fire INventory from NCAR (FINN, Wiedinmyer et al., 2011) in purple). Furthermore, the annual (July - June) GFED fire emissions are given with the right y-axis. Please note, that the right y-axis starts at 280 TgC per year for better visualization of the fire emissions.**

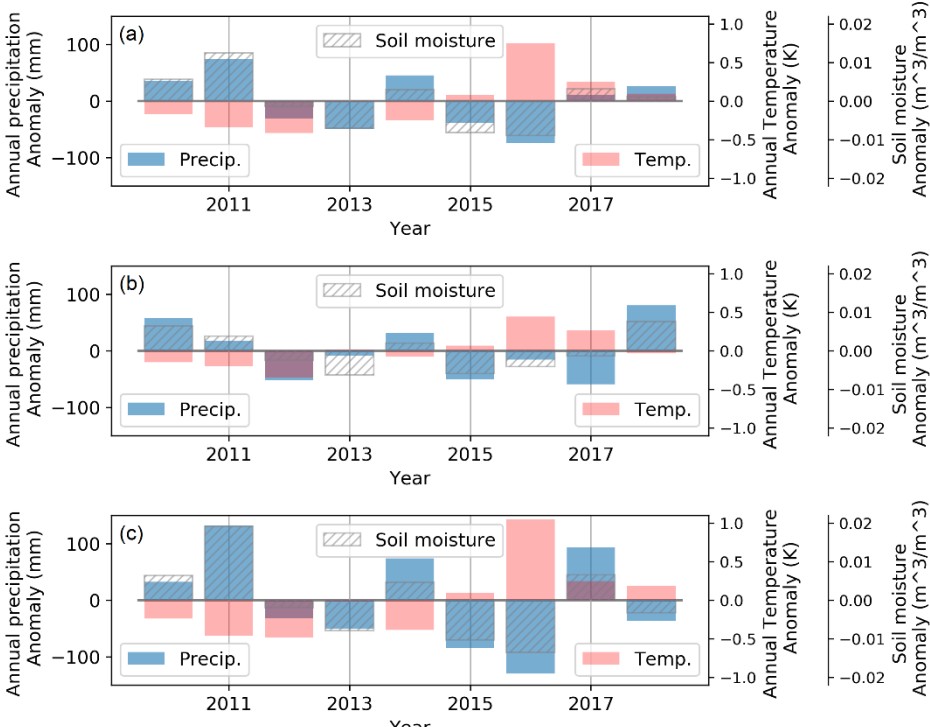

**Figure A11: Climate Anomalies. The annual anomalies of ERA5 precipitation, temperature and upper layer soil moisture are given in blue, red, and grey hashed. The annual anomalies are calculated by subtracting the individual long-term mean of the annual values and are given for the whole study region in panel (a), for the northern subregion in panel (b), and southern subregion in panel (c).**

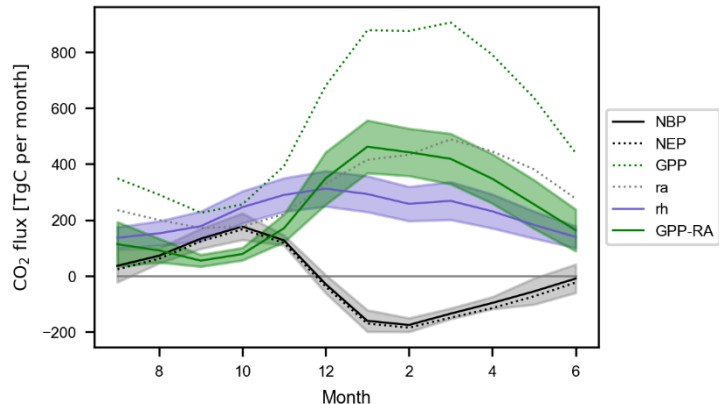

**Figure A12: Mean monthly CO₂ net and gross fluxes. Like Fig. 8 (a) but additionally with GPP and RA of the TRENDY selection.**

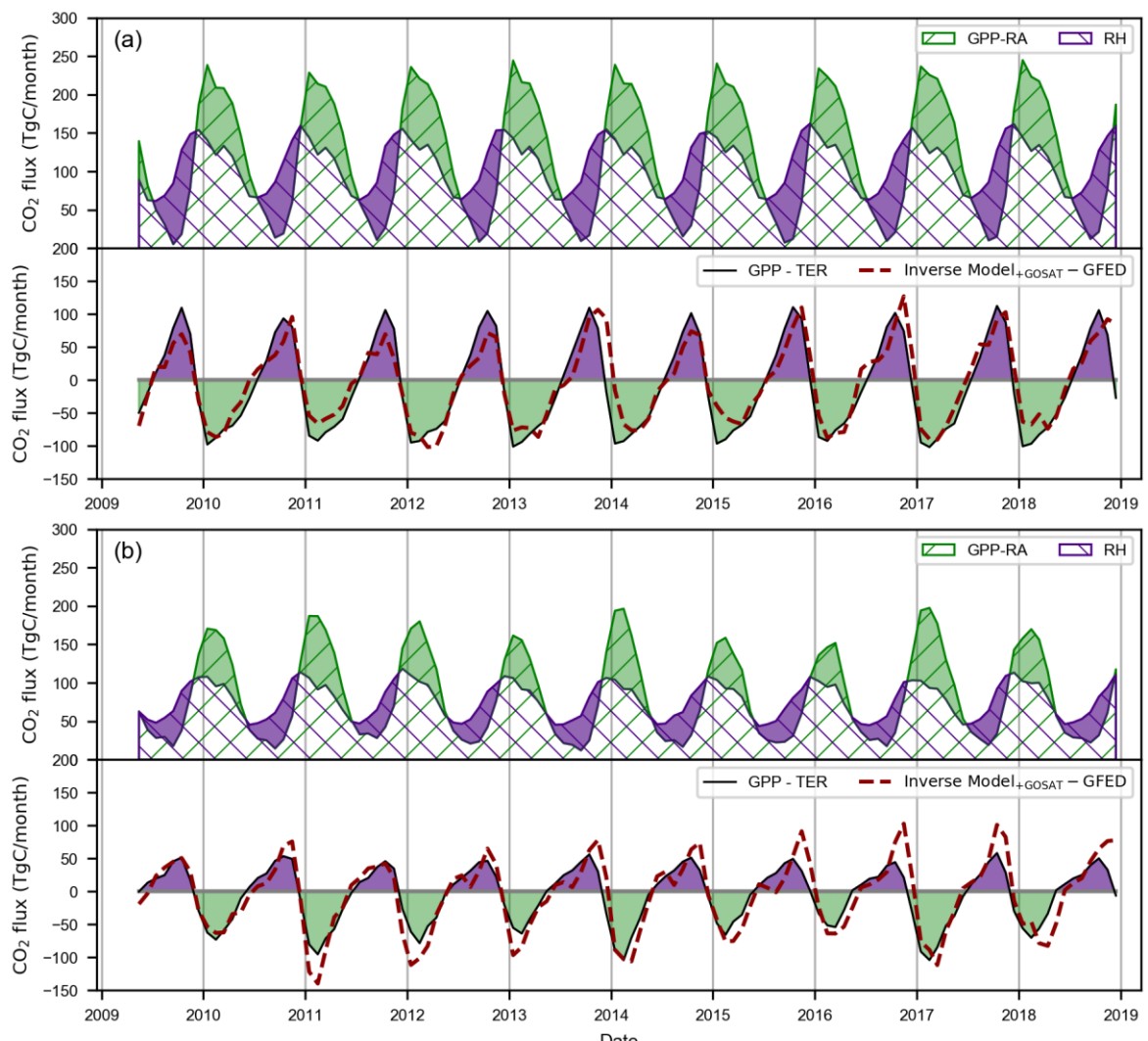

**Figure A13: Monthly CO₂ fluxes in northern (Panel (a)) and southern (Panel (b)) subregion. The monthly NEE, NPP (GPP-RA), and RH fluxes from the selected TRENDY models are given in black, green, and violet respectively for the northern southern African region in Panel (a). The TM5-4DVar/GOSAT+IS - GFED NEE fluxes are additionally shown in red dotted. The same is given in Panel (b) for the southern subregion.**

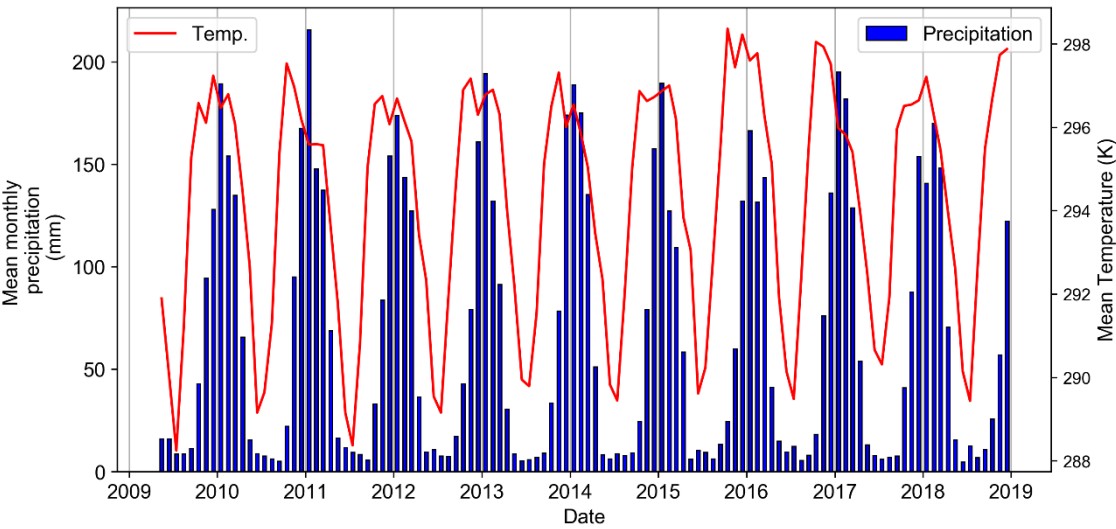

**Figure A14: Monthly precipitation and temperature as mean over southern Africa. The monthly precipitation is given as blue bars and the mean temperature as solid red line.**

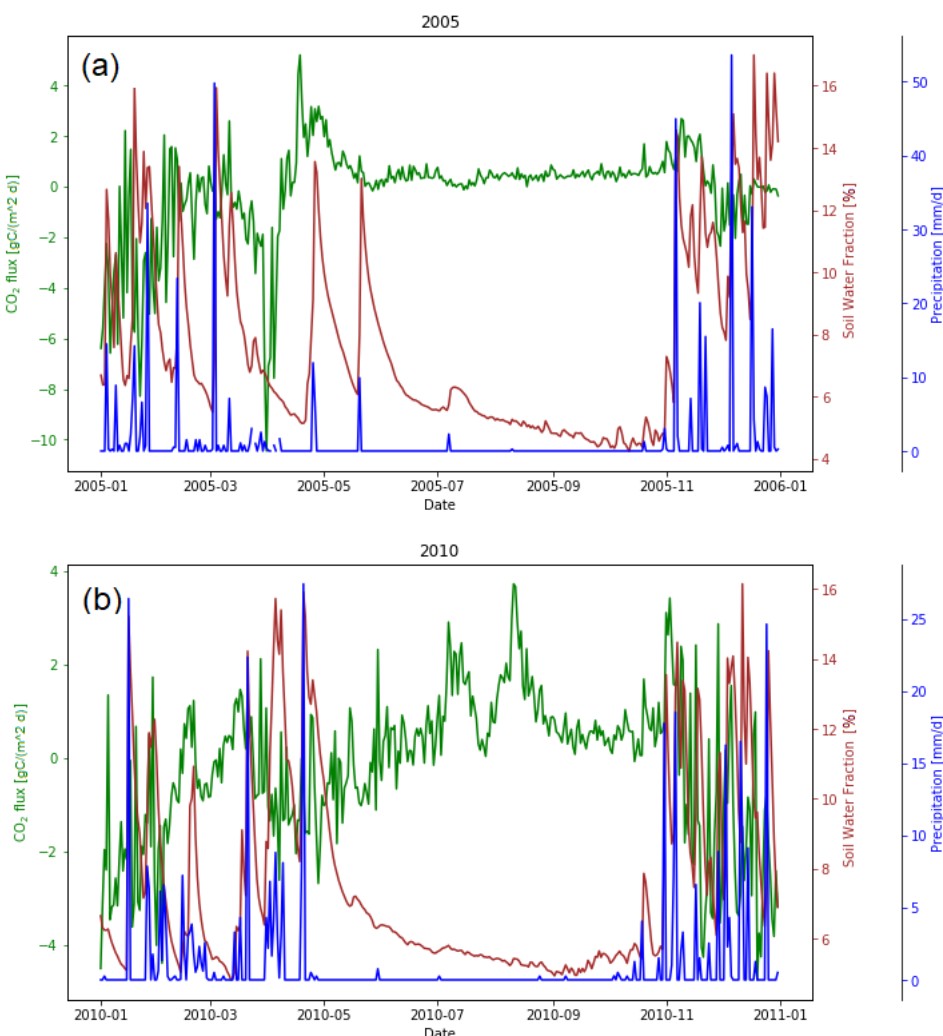

**Figure A15: Local data from FLUXNET eddy covariance flux tower in Kruger National Park. Daily mean net carbon fluxes (green), precipitation (blue) and soil moisture (red) measured by the FLUXNET station ZA-Kru (Archibald et al., 2009). Panel (a) shows the year 2005, Panel (b) shows 2010.**

Table A1: Summary of datasets

| Description | Dataset | Resolution | References |
|---|---|---|---|
| GOSAT XCO$_2$ | GOSAT/RemoTeC v2.4.0<br>GOSAT/ACOS v9r(Lite) | 10.5 km footprint<br>10.5 km footprint | Butz et al., 2011, 2022<br>Taylor et al., 2022, OCO-2 Science Team, 2019 |
| Validation XCO$_2$ | OCO-2 v11r<br><br>COCCON Gobabeb | 1.3×2.3 km footprint<br><br>local | Jacobs et al., 2024; OCO-2/OCO-3 Science Team, 2020<br>Frey et al., 2021, Dubravica et al., 2021 |
| Model XCO$_2$ based on in situ data | TM5 − 4DVAR/IS<br>CarbonTracker CT2022<br>CAMS v21r1 | 3°×2°, monthly<br>3°×2°, monthly<br>3.7°×1.81°, monthly | Basu et al., 2013<br>Peters et al., 2007; Jacobson et al., 2023<br>Chevallier et al., 2005, 2010, 2019 |
| In-situ-only inversions | TM5 − 4DVAR/IS<br>CarbonTracker CT2022<br>CAMS v20r1 | 3°×2°, monthly<br>1°×1°, monthly<br>3.7°×1.81°, monthly | Basu et al., 2013<br>Peters et al., 2007; Jacobson et al., 2023<br>Chevallier et al., 2005, 2010, 2019 |
| TM5-4DVar/GOSAT+IS | TM5-4DVar/RemoTeC+IS and TM5-4DVar/ACOS+IS | 3°x2°, monthly | Basu et al., 2013 |
| TM5-4DVar/OCO-2+IS | TM5-4DVar of MIP/LNLGIS | 1°x1°, monthly | Basu et al., 2013; Byrne et al., 2023 |
| MIP/OCO-2+IS<br>MIP/IS | MIP/LNLGIS experiment<br>MIP/IS experiment | 1°x1°, monthly | Byrne et al., 2023 |
| SIF | GOME-2 Daily_Averaged_SIF | 40 km x 40 km/80 km | Joiner et al., 2023 |
| FLUXCOM + GFED | FLUXCOMv1 NEE, RS_V006<br>GFED v4.1s | 0.08°×0.08°, 8-days<br>0.25°×0.25°, monthly | Tramontana et al., 2016; Jung et al., 2020<br>Van der Werf et al., 2017 |
| TRENDY$_{selection}$<br><br><br>TRENDY$_{others}$ | ORCHIDEE S3<br>ORCHIDEEv3 S3<br>CABLE-POP S3<br>YIBs S3<br>OCN S3<br>ORCHIDEE-CNP S3<br>JSBACH S3<br>CLASSIC S3<br>LPJ S3<br>CLM5.0 S3<br>DLEM S3<br>IBIS S3<br>ISAM S3<br>ISBA-CTRIP S3<br>JULES-ES-1.0 S3<br>LPX-Bern S3<br>SDGVM S3<br>VISIT S3 | 0.5°x0.5° [1]<br>2°x2° [1]<br>1°x1° [1]<br>1°x1° [1]<br>1°x1° [1]<br>2°x2° [1]<br>1.86°x1.88° [1]<br>2.80°x2.81° [1]<br>0.5°x0.5° [1]<br>0.94°x1.25° [1]<br>0.5°x0.5° [1]<br>1°x1° [1]<br>0.5°x0.5° [1]<br>1°x1° [1]<br>1.25°x1.88° [1]<br>0.5°x0.5° [1]<br>1°x1° [1]<br>0.5°x0.5° [1] | Krinner et al., 2005<br>Vuichard et al., 2019<br>Haverd et al., 2018<br>Yue and Unger, 2015<br>Zaehle et al., 2010<br>Goll et al., 2018<br>Reick et al., 2021<br>Melton et al., 2020<br>Poulter et al., 2011<br>Lawrence et al., 2019<br>Tian et al., 2015<br>Yuan et al., 2014<br>Meiyppan et al., 2015<br>Delire et al., 2020<br>Sellar et al., 2019<br>Lienert and Joos, 2018<br>Walker et al., 2017<br>Kato et al., 2013 |
| ERA5 meteorological data | ERA5-land data<br>total precipitation, upper layer soil moisture, temperature | 1°×1°, monthly | Muñoz Sabater 2019, 2021 |
| MODIS | MODIS (MCD12C1) data | 0.05°x0.05°, 2015 | Friedl and Sulla-Menashe, 2022 |

[1] all TRENDY model data is provided in monthly temporal resolution

The main characteristics and references of the observation and model data are listed. Links to the datasets are provided in the Data availability section.

Table A2: Monthly fluxes of TM5-4DVar/GOSAT+IS in southern Africa

| Year | Month | RT+IS | ACOS+IS | Mean | Year | Month | RT+IS | ACOS+IS | Mean |
|---|---|---|---|---|---|---|---|---|---|
| 2009 | 4 | -157.56 | -195.50 | -176.53 | 2014 | 3 | -218.74 | -194.84 | -206.79 |
| 2009 | 5 | -83.13 | -102.61 | -92.87 | 2014 | 4 | -160.54 | -153.89 | -157.21 |
| 2009 | 6 | 6.71 | 6.29 | 6.50 | 2014 | 5 | -84.72 | -81.25 | -82.99 |
| 2009 | 7 | 93.92 | 109.99 | 101.96 | 2014 | 6 | 30.42 | 42.46 | 36.44 |
| 2009 | 8 | 163.05 | 163.17 | 163.11 | 2014 | 7 | 82.04 | 99.66 | 90.85 |
| 2009 | 9 | 219.63 | 198.25 | 208.94 | 2014 | 8 | 95.93 | 122.13 | 109.03 |
| 2009 | 10 | 232.99 | 144.91 | 188.95 | 2014 | 9 | 215.17 | 154.74 | 184.96 |
| 2009 | 11 | 140.76 | 88.81 | 114.79 | 2014 | 10 | 229.27 | 176.57 | 202.92 |
| 2009 | 12 | -32.79 | -44.05 | -38.42 | 2014 | 11 | 199.23 | 168.02 | 183.62 |
| 2010 | 1 | -144.40 | -113.34 | -128.87 | 2014 | 12 | 36.93 | -35.25 | 0.84 |
| 2010 | 2 | -153.14 | -157.85 | -155.50 | 2015 | 1 | -73.64 | -86.37 | -80.01 |
| 2010 | 3 | -144.99 | -172.86 | -158.93 | 2015 | 2 | -139.19 | -135.31 | -137.25 |
| 2010 | 4 | -74.81 | -121.29 | -98.05 | 2015 | 3 | -153.79 | -149.43 | -151.61 |
| 2010 | 5 | -57.83 | -84.45 | -71.14 | 2015 | 4 | -144.28 | -131.81 | -138.04 |
| 2010 | 6 | 24.59 | 16.57 | 20.58 | 2015 | 5 | -62.78 | -63.61 | -63.19 |
| 2010 | 7 | 69.44 | 86.01 | 77.73 | 2015 | 6 | 2.16 | 22.31 | 12.24 |
| 2010 | 8 | 129.28 | 152.92 | 141.10 | 2015 | 7 | 49.88 | 85.39 | 67.64 |
| 2010 | 9 | 208.69 | 202.44 | 205.57 | 2015 | 8 | 117.11 | 107.91 | 112.51 |
| 2010 | 10 | 239.32 | 194.63 | 216.98 | 2015 | 9 | 189.95 | 139.90 | 164.93 |
| 2010 | 11 | 262.58 | 166.15 | 214.37 | 2015 | 10 | 225.03 | 150.79 | 187.91 |
| 2010 | 12 | 57.84 | -24.29 | 16.78 | 2015 | 11 | 259.19 | 212.22 | 235.70 |
| 2011 | 1 | -189.14 | -146.26 | -167.70 | 2015 | 12 | 112.16 | 78.85 | 95.50 |
| 2011 | 2 | -229.46 | -193.03 | -211.24 | 2016 | 1 | -72.92 | -69.47 | -71.20 |
| 2011 | 3 | -156.96 | -183.26 | -170.11 | 2016 | 2 | -148.67 | -155.69 | -152.18 |
| 2011 | 4 | -111.27 | -115.31 | -113.29 | 2016 | 3 | -176.60 | -134.03 | -155.32 |
| 2011 | 5 | -70.44 | -72.17 | -71.31 | 2016 | 4 | -159.32 | -128.91 | -144.11 |
| 2011 | 6 | 22.49 | 39.77 | 31.13 | 2016 | 5 | -77.83 | -56.86 | -67.35 |
| 2011 | 7 | 88.88 | 101.56 | 95.22 | 2016 | 6 | 28.77 | 72.38 | 50.58 |
| 2011 | 8 | 170.18 | 183.09 | 176.63 | 2016 | 7 | 61.68 | 117.42 | 89.55 |
| 2011 | 9 | 214.57 | 202.08 | 208.32 | 2016 | 8 | 111.76 | 166.74 | 139.25 |
| 2011 | 10 | 215.25 | 137.67 | 176.46 | 2016 | 9 | 178.65 | 176.21 | 177.43 |
| 2011 | 11 | 108.61 | 83.75 | 96.18 | 2016 | 10 | 278.49 | 178.25 | 228.37 |
| 2011 | 12 | -69.23 | -42.93 | -56.08 | 2016 | 11 | 344.93 | 213.55 | 279.24 |
| 2012 | 1 | -198.76 | -174.22 | -186.49 | 2016 | 12 | 126.39 | 48.90 | 87.64 |
| 2012 | 2 | -204.51 | -185.68 | -195.09 | 2017 | 1 | -141.60 | -144.98 | -143.29 |
| 2012 | 3 | -201.66 | -209.21 | -205.43 | 2017 | 2 | -218.16 | -157.23 | -187.70 |
| 2012 | 4 | -157.34 | -149.79 | -153.56 | 2017 | 3 | -266.37 | -195.15 | -230.76 |
| 2012 | 5 | -85.64 | -61.66 | -73.65 | 2017 | 4 | -171.98 | -145.48 | -158.73 |
| 2012 | 6 | 26.99 | 55.95 | 41.47 | 2017 | 5 | -87.55 | -94.62 | -91.09 |
| 2012 | 7 | 81.80 | 111.87 | 96.84 | 2017 | 6 | -4.45 | 17.30 | 6.43 |
| 2012 | 8 | 105.47 | 131.05 | 118.26 | 2017 | 7 | 36.00 | 108.33 | 72.17 |
| 2012 | 9 | 182.86 | 156.69 | 169.77 | 2017 | 8 | 125.62 | 175.62 | 150.62 |
| 2012 | 10 | 216.78 | 172.23 | 194.51 | 2017 | 9 | 191.89 | 212.30 | 202.10 |
| 2012 | 11 | 130.49 | 155.95 | 143.22 | 2017 | 10 | 285.32 | 197.40 | 241.36 |
| 2012 | 12 | -29.84 | -24.57 | -27.20 | 2017 | 11 | 233.14 | 175.95 | 204.54 |
| 2013 | 1 | -195.13 | -142.42 | -168.78 | 2017 | 12 | 3.21 | 3.05 | 3.13 |
| 2013 | 2 | -181.41 | -141.65 | -161.53 | 2018 | 1 | -131.45 | -111.65 | -121.55 |
| 2013 | 3 | -150.87 | -134.34 | -142.60 | 2018 | 2 | -119.89 | -127.09 | -123.49 |
| 2013 | 4 | -133.19 | -113.00 | -123.10 | 2018 | 3 | -167.60 | -135.00 | -151.30 |
| 2013 | 5 | -72.44 | -40.57 | -56.51 | 2018 | 4 | -208.14 | -153.04 | -180.59 |
| 2013 | 6 | 34.37 | 52.38 | 43.38 | 2018 | 5 | -137.36 | -102.90 | -120.13 |
| 2013 | 7 | 64.78 | 85.80 | 75.29 | 2018 | 6 | -21.20 | 23.47 | 1.14 |
| 2013 | 8 | 96.91 | 130.53 | 113.72 | 2018 | 7 | 29.86 | 98.30 | 64.08 |
| 2013 | 9 | 176.64 | 185.33 | 180.99 | 2018 | 8 | 110.99 | 163.25 | 137.12 |
| 2013 | 10 | 219.32 | 178.29 | 198.80 | 2018 | 9 | 202.02 | 201.28 | 201.65 |
| 2013 | 11 | 249.06 | 191.11 | 220.08 | 2018 | 10 | 182.51 | 179.17 | 180.84 |
| 2013 | 12 | 202.08 | 64.14 | 133.11 | 2018 | 11 | 223.74 | 184.91 | 204.33 |
| 2014 | 1 | -79.09 | -119.87 | -99.48 | 2018 | 12 | 226.30 | 148.33 | 187.31 |
| 2014 | 2 | -187.16 | -169.20 | -178.18 | | | | | |

The monthly fluxes of TM5-4DVar/RemoTeC+IS ('RT+IS'), TM5-4DVar/ACOS+IS (ACOS+IS), and the mean of both is given in TgC/month for the whole study region.

*Data availability*. GOSAT/RemoTeC2.4.0 XCO$_2$ data can be obtained from Zenodo https://doi.org/10.5281/zenodo.7648699 (Butz, 2022) (last access: 2024-05-15). GOSAT/ACOS data are available at https://oco2.gesdisc.eosdis.nasa.gov/data/GOSAT_TANSO_Level2/ACOS_L2_Lite_FP.9r/ (last access: 2020-07-28). OCO-2 data are available at https://disc.gsfc.nasa.gov/datasets/OCO2_L2_Standard_11r/summary (last access: 2023-06-28). CarbonTracker CT2022 CO$_2$ fluxes and concentrations can be downloaded from https://gml.noaa.gov/aftp/products/carbontracker/co2/CT2022/fluxes/monthly/ (last access: 2023-04-17) and https://gml.noaa.gov/aftp/products/carbontracker/co2/CT2022/molefractions/co2_total_monthly/ (last access: 2024-09-17), respectively. CAMS concentrations and fluxes can be found at https://ads.atmosphere.copernicus.eu/cdsapp#!/dataset/cams-global-greenhouse-gas-inversion (last access: 2021-10-07). GFAS emissions records are available at https://apps.ecmwf.int/datasets/data/cams-gfas/ (last access: 2020-11-13). CAMS and GFAS data were generated using Copernicus Atmosphere Service Information [2021], and neither the European Commission nor the European Centre for Medium-Range Weather Forecasts (ECMWF) is responsible for any use that may be made of the information it contains. The MIP data can be downloaded from https://www.gml.noaa.gov/ccgg/OCO2_v10mip/ (last access: 2022-05-06). GFED fire emissions are available at https://www.geo.vu.nl/~gwerf/GFED/GFED4/ (last access: 2020-07-10). FINN data were retrieved from the American National Center for Atmospheric Research https://www2.acom.ucar.edu/modeling/finn-fire-inventory-ncar (last access: 2020-11-18). ERA5-land data records contain modified Copernicus Atmosphere Service Information [2021] available at the Climate Data Store https://cds.climate.copernicus.eu/cdsapp#!/dataset/reanalysis-era5-land-monthly-means (last access: 2023-10-13). TRENDYv9 model output and FLUXCOM products are available at https://sites.exeter.ac.uk/trendy and http://fluxcom.org/CF-Download/, respectively. Data of the FLUXNET station ZA-Kru can be downloaded from the FLUXNET webpage: https://fluxnet.org/data/fluxnet2015-dataset/ (last access 21.11.2023). COCCON Gobabeb station data is available at https://secondary-data-archive.nilu.no/evdc/ftir/coccon/gobabeb/version2/ (last access: 2023-03-27). MODIS MCD12C1 data is available on https://search.earthdata.nasa.gov/search with the DOI 10.5067/MODIS/MCD12C1.061 (last access: 2024-03-24). L2 Daily Solar-Induced Fluorescence (SIF) from MetOp-A GOME-2 V2 data is available at from https://search.earthdata.nasa.gov/ (last access 2024-05-29). Monthly TM5-4DVAR data are available in Table A2.

*Code availability:* The code used in this study is available at https://zenodo.org/doi/10.5281/zenodo.12528504 or GitHub (https://github.com/ ATMO-IUP-UHEI/MetzEtAl2024)

*Competing interests*. The authors declare that they have no conflict of interest.

*Author contributions*. S.N.V., A.B. and E.-M.M. were involved in conceptualization and methodology. S.B. performed the dedicated TM5-4DVAR runs. E.-M.M. conducted the formal analysis and the visualization under the supervision of A.B.

and S.N.V. E.-M.M. wrote the original draft. All authors contributed to the interpretation of the results and the editing and review of the manuscript.

*Funding*. Funding for the developing RemoTeC was provided by the German Research Foundation (DFG) through grant BU2599/1-1, operations and data storage was supported by DFG grant INST 35/1503-1 FUGG. S.B. was supported by NASA grant 80NSSC20K0818.

*Acknowledgements*. We gratefully acknowledge the data storage service SDS@hd supported by the Ministry of Science,
Research and the Arts Baden-Württemberg (MWK) and the computing resources provided by the DKRZ under
project bb1170. E.-M.M. acknowledges a doctoral scholarship from the German National Academic Foundation. We thank the Japanese Aerospace Exploration Agency, the National Institute for Environmental Studies, and the Ministry of Environment for the GOSAT data and their continuous support as part of the Joint Research Agreement. We thank the OCO-2 science team for producing the GOSAT/ACOS L2 $XCO_2$ data. OCO-2 data were produced by the OCO-2 project at the Jet
Propulsion Laboratory, California Institute of Technology, and obtained from the OCO-2 data archive maintained at the NASA Goddard Earth Science Data and Information Services Center. CarbonTracker CT2022 results were provided by NOAA ESRL, Boulder, Colorado, USA, from the website at http://carbontracker.noaa.gov. We thank Stephen Sitch, Pierre Friedlingstein, and all modellers of the Trends in Net Land-Atmosphere Exchange project (TRENDY; https://blogs.exeter.ac.uk/trendy/). The long-term operation of the COCCON Gobabeb site in Namibia is supported by ESA
via the COCCON-PROCEEDS series of projects (contract no. 4000121212) and COCCON-OPERA (contract no. 4000140431/23/I-DT-Ir).

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
