# Peer review of "Seasonal and inter-annual variability of CO2 fluxes in southern Africa seen by GOSAT"

_EGUsphere, 2024_

## Referee Comment (RC2)

Review of Metz et al.: Seasonal and inter-annual variability of carbon fluxes in southern Africa seen by GOSAT, submitted to Biogeosciences

14 Oct 2024

**Review Summary**

The study by Metz et al. starts from satellite-retrieved XCO2 data analysis over a large southern African region dominated by semi-arid savannas and grasslands. They find a larger seasonal variability of XCO2 from GOSAT and OCO-2 compared to an ensemble of in-situ-optimized inverse models resampled at GOSAT soundings. They further point out that the discrepancy originates from the biospheric prior fluxes that control the optimized biospheric fluxes due to the lack of observational constraints. An understanding of the processes that affect the CO2 fluxes and their variability is gained through a comparison of the TRENDY v9 ensemble of Dynamic Global Vegetation Models (DGVMs).

The paper is well in the scope of Biogeosciences. The paper points out large discrepancies in DGVMs over semi-arid regions in southern Africa, and presents how satellite data can be used to evaluate these models. While the results of the paper do not include completely novel findings (like the authors already point out with appropriate references in the paper), the results are in agreement with previous studies carried out using other approaches, and should therefore be interesting to the readers of the journal. The paper is particularly well written, and the methods and results are clearly presented. I also like how the paper advances from one result to the next one in a way that is pleasant to follow. I recommend publishing the paper in Biogeosciences after the authors have considered the constructive comments detailed below. I have also gone through the comments from Reviewer 1 and try not to repeat their message, although some of my comments might be of similar nature.

**Major comments**

1) I don't think the title of the paper is fully descriptive of its contents. Could the title be for instance along these lines: "The potential of CO2 satellite observations to inform and evaluate modelled carbon dioxide fluxes in Southern Africa" or something similar. At least, I would propose to change "carbon" to "carbon dioxide" or "CO2" because methane and CO are not discussed in the paper.

2) The study region is very large and the CO2 fluxes can be quite heterogenous within the study region (for example, on NOAA's website one can see that CarbonTracker concludes part of the study region to be a net source and part a net sink of CO2; https://gml.noaa.gov/ccgg/carbontracker/fluxmaps.php?region=afr&average=annual#imagetable and even though CT is not the best-performing model in the

region as the authors point out, this underlines the heterogeneity of the region and makes one wonder how much you can average over without losing important information). I think that the averaged XCO2 time series (Fig. 2) is to some extent dependent on the sampling. The number of GOSAT observations likely varies from only few tens to some 1000-2000 per month over this region, depending on the season. It would be good to show the time series of the number of GOSAT observations (both products) for example added in Fig. 2 and also a map of the spatial sampling density for example around the XCO2 minimum and XCO2 maximum (this could be in the Appendix), and discuss how the sampling affects the results of the concentration-based analysis and the inverse modelling results.

**Minor comments**

Abstract: I suggest highlighting the discovered role of grassland carbon uptake for IAV which I consider as a key result of the work. It would also be helpful to give key numbers of TgC so that the reader can contrast those more easily.
Line 46: Spell Orbiting Carbon Observatory -2 with a hyphen
Lines 56-57: CO2 exchange fluxes → CO2 exchange
Line 57: to flux estimates → to those (to avoid repetition)
Line 69: Valentinti → Valentini (check!)
Line 69: grass land → grassland
Figure 1: This is a massive study region. To make this clearer to the reader, I suggest adding a scale bar representing for example 200 or 500 km or similar. In addition, please add the locations of the Gobabeb COCCON and the Kruger National Park measurement with pins or equivalent symbols.
Line 76: I believe that the description of the timing of the fire season applies to the study region in question and not the whole Africa. Please specify in the text.
Section 2.2: Did you also include satellite data over the ocean in your analysis?
Line 82: column-average → column-averaged
Line 85: How different is RemoTeC v2.4.0 compared to RemoTeC v2.3.8? Please summarise the key developments after v2.3.8 in the text.
Lines 87-90: I understand that both products have been used as two separate products (e.g., apply their own quality filtering and potential bias corrections). Did you compare ACOS and RemoTeC by considering only the exact same GOSAT soundings? If so, what did you learn about the product XCO2 differences? While this question may not be in the core of the paper, I think it is of high interest to learn as much as possible about the satellite observations' differences in particular over these regions that are poorly sampled by any other measurements. In addition, the differences in the satellite products can result in flux differences of several tens TgC per month for this region (Fig. 3), which is significant and interesting. Depending on the authors' view, this might even merit a sentence in the Conclusions to address the need of validation measurements in this region to be able to reconcile residual differences in the satellite products.
Lines 87-90: In Fig. 2, we are seeing differences in the sub-ppm scale. It would be helpful to give guidance (with references) to the (global) accuracy and precision of

these GOSAT products in this part of the text to help the reader understand to what extent we can trust the satellite observations.

Line 91: I think it would be more descriptive to say "evaluation" instead of "validation".

Line 94: The COCCON data were not co-located with the satellite observations, right? Did you do any filtering to the data, for example in the early or late hours with large solar zenith angles? Please specify in the text.

Sect. 2.3.1: The three inversion models also use different data assimilation schemes, right (4DVar vs. enKf)? This could be specified in the text.

Sect. 2.3.1.: Did you assimilate also GOSAT observations over oceans? Please specify.

Sect. 2.3.1.: Question, out of curiosity: did you also consider a GOSAT-only inversion with no in-situ data?

Line 113 (and elsewhere): please use \citep[TM5-4DVar,][]{Basu et al reference} LaTeX command to get the rid of the extra parentheses within parentheses (if working with LaTeX).

Lines 141-144: Did you apply the averaging-kernel correction when comparing the cosampled model concentrations to satellite data? The effect is not likely to be large but it might be non-negligible since the differences in question are mostly in the sub-ppm scale.

Line 156: Does the FLUXCOM have a product version that could be referenced here? We have recently worked with a FLUXCOM product that had a strong positive bias in $CO_2$ fluxes over both Northern and Southern Africa. This seems to not be the case with the product in the paper. In fact, if I interpret Fig. 3 correctly, the FLUXCOM NEE would give out either close to zero or negative values without GFED fire emissions, correct?

Line 173: Methods → Data and Methods

Lines 176-178: This reminds me of similar GOSAT-to-model discrepancies that were pointed out by Lindqvist et al. (ACP, 2015; https://acp.copernicus.org/articles/15/13023/2015/) and their reasoning, albeit with much earlier versions of the data and models.

Figure 2: Please consider adding the monthly data density or, if the result is too crowded, presenting the data density time series using other means.

Figure 2: What does the CS in the subscript of the "Inverse models" stand for?

Line 190: e. g. → e.g.,

Sect. 3.2: I think it is interesting and worth pointing out how you can already "see" the impact of the fluxes in the detrended concentrations (Fig. 2). The well-aligned timings also suggest that the concentration maxima and minima are driven by the fluxes from this region as opposed to transport from elsewhere. This suggests that satellite data can already in its concentration form have potential and be useful for an evaluation of models in a region (even without inverse modelling!).

Figure 3 (a): the legend is partly on top of the plotted data. Please revise such that the data are fully visible. This comment applies to some subsequent figures as well.

Line 204: remove "methods"

Line 244: a → an

Line 253: processed based → process-based

Sect. 3.3.: I understand that if the focus of the paper is to learn about the processes affecting the large-scale $CO_2$ exchange in this region, it is meaningful to focus on the DGVMs that agree with the GOSAT inversions. However, as also Reviewer 1 points out, it seems like a missed opportunity to not discuss the models that agree the worst. I also

wonder how generalizable these findings might be for other semi-arid regions in the world; one good example already published by the authors about the Australian fluxes (Metz et al., 2023). I get the feeling that the authors have already learned much more from individual model performance than what they are writing in the paper. I know it is not a preferred task for anyone to point out flaws in someone else's product but this is sometimes necessary to enhance openness that might further advance science.

Figure 6: The shading mentioned in the caption is not visible (at least in my printed version). The legend covers part of the data, please revise the figure. Why is GOME-2 SIF used instead of GOSAT SIF or OCO-2 SIF? I wonder how different these would be.

Lines 278-279: CAMS cannot be identified from the spread of models in Figs. 2-3.

Lines 283-284: You could add a very recent reference in your discussion of fire emissions in the regions. This one also seems to suggest underestimated fire emissions. Van der Velde et al. (2024): https://agupubs.onlinelibrary.wiley.com/doi/full/10.1029/2023GL106122

Lines 300 and 304: I think the correct spelling is La Niña and El Niño

Line 302: "enhanced surface near soil moisture" this sentence might be missing a word?

Sect. 3.4 title: I'm not sure what is meant by gross fluxes. Could you just say fluxes?

Line 321: remove "fluxes"

Line 344: Suggest to rewrite the sentence starting "A by 1-2 months…" as it is not easy to read.

Lines 359-362: To me, carbon uptake by grasslands dominating the IAV of the entire southern African fluxes sounds like an important finding that should be brought to the Abstract and perhaps even contrasted in magnitude against fire emissions to the reader so that the magnitude is easier to grasp. At least to me this finding highlights the importance of an improved understanding (and modelling) of the sensitivity of grasslands.

Lines 359-362: This is an example where the sampling of the satellite data for GOSAT inversions might end up affecting the resulting reasoning (see my Major comment #2).

Line 374: rain induces → should this read rain-induced?

Figure A1 caption: und → and

Figure A1 legend: what does "CS" stand for? Also, the legend is covering part of the data, please revise.

Figure A3: it seems that months 10 and 11 are particularly challenging. Could you please add a bit of discussion related to this in the text?

Figure A8: Just a comment: this amount of precipitation is likely to affect the number of satellite observations of the region. Please see my Major comment #2.

Figure A9: Are these measurements assimilated in any of the in situ optimized inversion models discussed in the paper? The time span would match part of the time period discussed in the paper.

---

## Author Response (AR1)

Author's response

We thank the reviewers of our manuscript for the constructive and detailed comments. The comments helped us improve our manuscript. We address them 1 by 1 below. Our response is given in red italic. Changes which were applied in the manuscript are given in red and plain text. These updates have no impact on the conclusions of the paper.

As asked by the precedent review file validation, we revised the color scheme of our figures to improve the interpretability for readers with color vision deficiencies. Furthermore, we needed to update the CT2022 $CO_2$ concentrations (Fig. 2, Fig. A1-A4), as the previously downloaded data was erroneous. The update caused only minor changes in the monthly CT2022 $CO_2$ concentrations and has no impact on any conclusions in our manuscript.
* * *
Response to review 1 of "Seasonal and inter-annual variability of carbon fluxes in southern Africa seen by GOSAT"

Review: Seasonal and inter-annual variability of carbon fluxes in southern Africa seen by GOSAT

by Eva-Marie Metz, et al.

September 23, 2024

**1 Review Summary**

In this work, the authors use atmospheric inversion systems, Dynamic Global Vegetation Models (DGVMs) from the TRENDY suite, satellite observations of total column carbon dioxide ($XCO_2$), satellite observations of SIF, various ground based observations, and a few other ancillary sources of information to probe questions about the terrestrial carbon cycle (TCC) of southern Africa (SA), which is largely dry grasslands and savannas. The results seem to be three fold: (1) some DGVMs do not correctly parameterize the terrestrial carbon cycle in southern Africa, while some select models do tend to agree with flux inversion results that assimilate satellite observations, (2) the satellite observations of $XCO_2$ (mainly from GOSAT, but by extension to other sensors like OCO-2 and OCO-3) are very useful to constrain the terrestrial carbon cycle where highly precise ground-based in-situ measurements are not available (which is pretty much everywhere in the tropical and southern hemisphere lands!), and finally, (3) the inter-annual variability (IAV) of southern Africa is driven mainly by variability in the photosynthetic uptake of $CO_2$, while the seasonal cycle is influenced by enhanced soil respiration at the onset of the predictable and distinct rainy season.

To my somewhat limited understanding of DGVMs and flux inversions, the technical aspect of the work appears to be very solid. The satellite data seem to be used properly. It is clear

that a tremendous amount of time and effort was spent on this research, as the number of models and data sets involved is quite impressive. I have no doubt that many of the results and conclusions will be found useful and important by the science community.

My primary, overarching constructive criticism about this work is that to me there seems to be some missed opportunities to better capture the multi-faceted conclusions that I stated above. The tone of the paper is a bit technical, with some of the findings left as implied ideas, rather than explicit recommendations. The topic of the paper seems to be a bit unsure: is it about the limitations of some DGVMs?, or about the utility of the satellite observations?, or about the terrestrial carbon cycle in southern Africa? I think the answer is that all three components are of interest,which is really cool! I feel like the authors could improve the message drastically by spending a little time developing some of these conclusions into more pronounced take-home messages. To offset any increased verbiage in the main body of the text, the authors could shift some of the more technical aspects into the Appendix. To me that would work well and increase the value of the paper to the scientific community! Of course, this idea is optional as I understand the limitations of time and resources.

*Thank you for your feedback. Indeed, our manuscript has multiple facets. Taking into account the comments of you and reviewer 2, we developed the conclusions regarding the advantages of satellite data, the DGVM analysis, the MIP model ensemble, and the inter annual variability further. We will add additional supporting figures in the appendix and we will include concluding statements in the main text. Please find these changes below in our answers to your major and minor comments.*

The paper is well written in clear English, the organization of the material is logical. The subject matter is highly appropriate for the Copernicus journal Biogeosciences. My overall recommendation is to publish with major or minor revisions, depending on the author's desire to increase the utility of the paper. It could be published with only minor corrections as outlined below. Actionable items are given below.

**2 Major Concerns**

- There seems to be some disjoint between the chosen title of the paper and the content discussed within. For example, couldn't the title just as well be: "Informing DGVMs by use of satellite-based observations of $CO_2$: case study over southern Africa"? This goes back to my summary points that the paper seems to have several facets but it is a bit unclear to me which one is the key focus.

  *We agree that the chosen title is universal. As you pointed out, our manuscript has multiple focus areas. It describes methodological advancements, limitations and opportunities of datasets, and findings about the African carbon cycle. For this reason, we hesitate to use a more specific title and prefer*

*to keep the current title (with adaptations from reviewer 2: "Seasonal and inter-annual variability of $CO_2$ fluxes in southern Africa seen by GOSAT"). For example, in the title suggested by you the identification of processes driving the African carbon cycle is missing.*

- How can this work inform the community about the "bad" DGVMs? What is different about those models as compared to the "good" ones (the ones that more closely match the satellite based inversion results). That seems like a missed opportunity to me. I have to admit that I am not very familiar with DGVMs. I provided some specific questions in the Minor Comments section.
  *Taking into account your remarks in the minor comments, there are two aspects we will stress further in our revised manuscript.*
  1) *Only a few DGVMs are found to agree with the satellite based fluxes. So the majority of models shows large uncertainties in the African fluxes.*
  2) *We identified the dephasing of GPP and rh to be a prerequisite for DGVMs to accurately represent the Southern African carbon fluxes. We will elaborate which processes might cause the dephasing and need to be implemented in the DGVMs.*

  *Please see the response to the minor comments for more details and the planned changes in the manuscript.*

- I would like to see a bit of material indicating (a) spatial and temporal coverage of the satellite observations, and (b) a rudimentary comparison of the RemoTec vs ACOS $XCO_2$. It just states that a mean value of the two is used. It would be easy to put this material into the Appendix to avoid the main document from beccoming too long and detailed.
  *We will include three new figures into the Appendix:*
  1) *Maps of the number of satellite measurements for the first half and second half of the year and for the months of the largest emissions (Sept.-Nov.)*
  2) *Timeseries of the number of satellite measurements for GOSAT/ACOS, GOSAT/RemoTeC, and OCO-2.*
  3) *Monthly timeseries of GOSAT/ACOS and GOSAT/RemoTeC only for the intersection of GOSAT soundings included in both data products.*
  *Please find the figures (A5, A6 and A1) and a detailed discussion in the response to your minor comments below.*

- SIF data: Please clarify why a GOME SIF product was selected over GOSAT? Also, isn't the general statement "SIF correlates with GPP" fraught with debate in the science community? Isn't the relationship often/sometimes non-linear as a function of biome type? Perhaps a few additional citations are warranted?

If the SIF-GPP relationship were non-linear for the grasslands and savannas, how would that potentially impact the science conclusions?

*GOSAT SIF has a much sparser sampling than GOME SIF. As we compare the SIF against monthly GPP given by the vegetation models for the whole study region, GOME-SIF is better suited.*

*The correlation of SIF and GPP depends on the spatial and temporal resolution of the data. Monthly SIF and GPP data on a regional scale was shown to have a close to linear correlation (Pierrat et al., 2022, Zhang et al., 2016A, Pickering et al., 2022, Zhang et al., 2016B). We will add the citations in the main text. Moreover, it is important to note that our analysis does not depend on a strictly linear relationship. We only assume SIF to increase with GPP, which is not under debate. In our response to your minor comments (please see Fig. RC1_1 and the describing text nearby), we furthermore show that our SIF analysis is independent from biome specific relationships of SIF and GPP.*

- Section 3.2 Top-down and bottom-up $CO_2$ fluxes: A bit more detail might be needed about the selection of the 3 models discussed around L229-230. Why do other models (that don't represent well the TM5-GOSAT inversions) stick closer to the prior? Is it simply a matter of loosening the prior covariance in those models? What else can be said about the "bad" models?

*The MIP dataset provides prior and posterior fluxes and posterior concentration co-samples on OCO-2 $XCO_2$ measurements. Using this data, we made the following observations for the months of high emissions (Sept.-Nov.):*

1) *Assimilating OCO-2 most of the time but not always increases the $CO_2$ fluxes and reduces the difference to GOSAT inversion.*
2) *The 'good' MIP models reproduce the OCO-2 measurements best.*
3) *Most of 'bad' MIP models stick closer to the prior for the emission phase from September to November.*

*We will describe and discuss the conclusions we draw from the MIP data more explicitly in a revision of our manuscript. To do so, we will modify Fig. A3 (then Fig. A7), include two additional figures, and add a discussion in the appendix (text A1) as follows below. The MIP dataset does not provide data on the error assumptions and covariances used. Furthermore, the MIP models do not use a common prior flux but have individual assumptions. Given that, we cannot analyze the findings above in more detail e.g. it is not possible to analyze why the 'bad' models tend to follow the prior more closely.*

[revised manuscript text omitted]

- TRENDY is an important aspect of this research. Recommend providing a sentence or two background about the project and it's relevance. Also, I don't think the acronym is defined!

  *We will add the following modified sentences to the methods section of the manuscript:*

  *"We compare the top-down $CO_2$ fluxes to bottom-up flux datasets from DGVMs as collected by* **version 9 of the intercomparison project "trends and drivers of the regional-scale sources and sinks of carbon dioxide (TRENDY, Le Quéré et al., 2013). The project was established to support the annual global carbon budget estimation conducted by the Global Carbon Project (e.g. Friedlingstein et al., 2020)**.*"*

- I think it's worth making a statement along the lines of "whatever we can do with the long GOSAT record, we will be able to do with a long OCO record, as well as future CO2 sensors too"! That is, the results are not specific to GOSAT (except that it has 2009-2014 measurements that were not observed by OCO); the results are more general in that any satellite-based observation of CO2 are going to inform the TCC.

  *We will include the following concluding sentences in Section 3.2. when discussing the importance of satellite measurements in atmospheric inversions:*

  *"Satellite $CO_2$ concentration measurements, therefore, provide a unique information source and are especially valuable in regions with sparse in situ measurement coverage. The already long record provided by GOSAT will be more and more complemented over time by the growing record of OCO-2 and future $CO_2$ sensors providing even more extensive measurements."*

**3 Minor Comments**

- Line 18: Recommend some rewording to "…differences between atmospheric inversions performed on satellite based observations versus inversions that assimilate only in situ measurements." or similar.

  *We will reword the sentence according to your suggestions.*

- Line 19. Break the two sentence apart. "…in situ measurement. This suggests limited…"

  *We will do that.*

- Line 21. Define TRENDY.

  *We will add the definition of TRENDY as described above in our response to your major comment.*

- Line 22-26: Additional emphasis needed as to whether this is a substantial new finding, or just follows expectation.

  *We will adapt the sentence in the abstract:*

*"Doing so, our satellite-based process analyses pinpoint photosynthetic uptake in the southern grasslands to be the main driver of inter-annual variability of the southern African carbon fluxes, agreeing with former studies based on vegetation models alone."*

- Line 28: replace "slows down" with "mitigates" or similar.

  *We will do that.*

- Line 48: Flattered, but I don't think Taylor, 2022 is appropriate in this context.

  *We included the citation, because the seasonal variability of the $CO_2$ concentration differences between GOSAT/ACOS and in-situ-based models over southern Africa is clearly shown in Figure 11 of your paper. However, this is nothing which is mentioned explicitly anywhere in the text. We will remove the citation.*

- Line 57: avoid using "fluxes" twice. Maybe replace the first instance with "results".

  *We will do that.*

- Line 58: The phrase "machine learning approaches" seems a little misleading in this context as the science world is currently being inundated with ML science, but the ML here is a secondary or tertiary issue. Recommend just saying "FLUXCOM products" instead of "machine learning approaches".

  *We will replace 'machine learning approaches' by 'FLUXCOM product'.*

- Lines 58-60: But what conclusions or statements can be made about the "bad" DGVMs? What is different about "good" models? That could potentially increase the utility and scope of the paper?

  *In general, it is difficult to compare the individual TRENDY models with respect to the implemented processes because the DGVMs differ largely in the included processes, the process implementations and the assumed conditions (land cover, soil type distribution…). Furthermore, disentangling the processes which dominate the $CO_2$ exchange on regional scale is challenging. In our paper, we succeed in identifying one key process: the dephasing between vegetation fluxes and heterotrophic respiration. Focusing on that and also taking into account the comment of reviewer 2, we will add the following paragraph in the discussion of the DGVMs gross fluxes:*

  "It is noteworthy that large parts of the not selected, 'other' TRENDY models miss the dephasing between RH and GPP-RA. Their NBP estimates, therefore, do not agree with the emissions around October found by the satellite inversion. Implementing soil respiration due to rewetting more accurately in those models could improve their agreement with the satellite-based fluxes. Metz et al. (2023) found that the dephasing in the TRENDY models is most likely caused by a different response time of soil respiration and vegetation growth on precipitation e.g. water needs to percolate into the deeper soil

layers with plant roots to initiate plant growth, whereas heterotrophic respiration is driven by upper soil layer soil moisture or precipitation. The implementation of such a time lag between heterotrophic respiration and GPP seems to be a necessary but not sufficient prerequisite to accurately capture the seasonal carbon flux variability in semi-arid southern Africa. Our results call for studies on how to implement the response of ecosystems on soil rewetting more accurately to improve the consistency and accuracy of the TRENDY ensemble in semi-arid regions."

- Line 63: In the text and on the Fig 1 map, Madagascar is listed and included in the sutdy area. But the study area is further broken into north and south, excluding Madagascar. Is Madagascar actually used anywhere in the study? Maybe it should be removed for clarity?

  *Madagascar is included in the big study area as indicated in the map to investigate the $CO_2$ flux of the whole African continent < -10°S. When breaking down Africa in North and South we excluded Madagascar. Its climatic conditions differ from the mainland, and it was difficult to assign it to either the northern or southern region. Including Madagascar in the subdivision by cutting it at 10°S, does not change the results for the northern and southern region as the signal is dominated by the mainland.*
  *We will clarify the handling of Madagascar in the caption of Figure 1:*
  "Madagascar is part of the main region, but it is excluded in the subdivision."

- Fig 1: looking at the biome types, I'm left wondering why -5 degrees latitude was not used as the northern boundary rather than -10 degrees? Looks like there is a lot of the same vegetation type between -5 and -10, namely the orange color (savannas)? Maybe the exact latitude range was defined in one of the referenced studies given in this section.

  *We defined our region in agreement with the region definition in Mengistu and Mengistu Tsidu (2020) as stated in line 64.*

- Fig 1: Could be useful to make a companion figure (a second panel of the same figure) with only 7 colors and % type per full region and per sub region. Seven relevant biomes would be grasslands, savannas, woody savannas, open shrublands, closed shrublands, water, and everything else combined as "other". Then a slightly different, more distinct color separation could be used for the types. Currently many of the colors run together. This is a minor point and probably not worth the effort that it might take!!

  *We added the suggested second panel. In the second panel, we coarsened the resolution to 1x1° to visualize the common resolution of the used data. Furthermore, we used less colors (5, for savanna, woody savanna, grassland, shrub, and others). Only a small minority of pixels is classified as closed shrub and inland water. Therefore, we included them in 'shrub' and 'other', respectively. We included pie charts to visualize the shares of the individual*

*land cover classes in the main and the subregions. Below, you find the revised Figure 1, which includes also changes in response to reviewer 2's suggestions.*

[Figure]

Figure 1: Study region southern Africa. The land cover in the study region is given based on MODIS (MCD12C1) data (Friedl and Sulla-Menashe, 2022). Additionally, the main region used for the analyses is depicted as a red box. In the inlet map on the right side, the land cover is aggregated in larger land cover classes and on a 1°x1° spatial resolution, which is used for most of the analyzed data. The main region, thereby, comprises 547 grid cells. The dashed boxes show the subdivision into a northern and southern region. Madagascar is part of the main region, but it is excluded in the subdivision. The pie charts depict the share of the different land cover classes in the main study region (M), the northern subregion (N), and the southern subregion (S). The locations of the COCCON measurement site Gobabeb (Frey et al., 2021; Dubravica et al., 2021) and the flux tower in Kruger National Park (Archibald et al., 2009) are given as red circle and diamond.

- Section 2.2 Total column CO2 measurements:
  - I'd like to see some basic plots of satellite coverage. Maps of densities, and time series of densities for the three products (GOSAT RemoTec, GOSAT ACOS, OCO-2) would be helpful.
  *We will include the density maps and the time series in the Appendix and include a new sentence in the main text.*
  *"The number of GOSAT measurements (see Fig. A5 and Fig. A6) is variable throughout the year with the smallest numbers occurring during the rainy season around December and January."*

[Figure]

Figure A5: Number and distribution of satellite $CO_2$ concentration measurements above southern Africa. (a), (d), and (g) Total number of GOSAT/ACOS, (b), (e), and (h) GOSAT/RemoTeC, and (c), (f), and (i) OCO-2 data per 3°x2° grid cell for (a) - (c) the months of carbon uptake (January – June), (d) - (f) the emission season (July – December), and (g) – (i) the month with the strongest emissions. GOSAT/ACOS and GOSAT/RemoTeC measurements from 2009 to 2018 and OCO-2 measurements from 09/2014 to 2018 are included. The maximum of the color scale is the same for all time periods, but different for OCO-2 than for GOSAT/ACOS and GOSAT/RemoTeC. Compared to GOSAT/ACOS, GOSAT/RemoTeC has a reduced number of measurements, as RemoTeC applies stricter filtering of the GOSAT soundings.

[Figure]

Figure A6: Number of satellite measurements per month. The amount of satellite measurements of the GOSAT/ACOS (red dashed), GOSAT/RemoTeC (dark red solid), and OCO-2 (grey dotted) dataset are given. Note that the number of OCO-2 measurements is shown divided by 100 to enable a comparison to the much less abundant GOSAT measurements.

- ○ Line 88: It would be a good place to drop the Taylor, 2022 citation.
  *We will do that.*
- ○ Line 88: The coverage period for ACOS GOSAT v9 is 04/2009 - 06/2020.

*We will adapt that.*

- ○ Line 91: "…version X (vX) XCO2…" Probably version 10 OCO-2 data were used? But maybe v11? The brand newest version is now up to v11.2, although difference between v11 and v11.2 will most likely not be significant for the research here. If v10 was used, please cite [Taylor, AMT, 2023]. If v11 was used, please cite [Jacobs, AMT, 2024].
  *We use OCO-2v11.1r. Thank you for the information about the citation. We will include Jacobs 2024 in the manuscript.*

- ○ Line 91: Probably OCO-2 Land-Nadir and Land-Glint (LNLG) observations were used? Please specify.
  *Yes, we will specify this in the revised manuscript.*

- ○ Line 101: Is it fair to describe this as a piece-wise linear correction?
  *In Line 101, we describe how we calculate the background, which we subtract from the $CO_2$ concentration to obtain the seasonal variability in the study region. The background subtraction is only used in Fig.2 and in the appendix to visualize more clearly the seasonal changes (e.g. time of maximum or minimum concentration) and the differences between satellite and in-situ-based models. The conclusions drawn from this figure are independent of the chosen background. Most importantly, we do not use the background subtraction in the TM5-4DVar/GOSAT+IS inversion, but the total, not detrended $XCO_2$ measurements. We added a sentence in the methods section 2.3.1 to clarify that:*
  "We use the individual total $CO_2$ concentration measurements, i.e. we do not apply any detrending or spatiotemporal averaging. Detrending and spatiotemporal averaging is only applied for visualization purposes to show the variability in the monthly $CO_2$ concentrations (Section 3.1)."
  *Hence, we think that a piece-wise linear background is fully sufficient for the illustrative purpose we use it for.*

- ○ Line 97: Not sure if it is perfectly relevant, but might be worth mentioning, (a) the calculation of atmospheric growth rate described in Appendix A of [Taylor, AMT, 2023]. See Fig. A3., and/or (b) the new work published by [Pandey, AGU Advances, 2024].
  *Thank you., we will include that:*
  "The growth rates are … and their calculation is further described in Taylor et al. (2023, Figure A3) and Pandey et al. (2024)."

- Section 2.3 Fluxes
  - ○ *We will implement the next eight minor comment when revising our manuscript*
  - ○ Line 117: Remove "Thereby,"

- Line 120: "Furthermore, the inversion systems…" (careful with the generic "models" as there are many models of different types used in this work!).
- Line 125: Replace "fed into" with "assimilated".
- Line 130: Missing "the" before "MIP".
- Line 131: Missing "by" between "fluxes" and "assimilating".
- Line 131: Change "satellite CO2" to "satellite XCO2".
- Line 131: Change "data together with" to "observations together with".
- Line 131: Remove "Thereby," and begin sentence with "All MIP…".
- Line 133: Was "LNLGIS" previously defined?

  *No, thank you. We will define it.*

- Lines 137: Do all the atmospheric inversions impose the same anthropogenic fossil emissions and fires emissions? I can't remember if that was a common constraint on the OCO MIP or not. If no, then couldn't differences in those constraints have impacts on your results since you are interested in NBP estimates? Maybe it does not matter? I'm not sure. Just seems like maybe one of those things that starts falling apart if you start poking around the edges?

  *All models in OCO MIP use the same fossil emissions (Byrne et al. 2023). However, they differ in the imposed fire emissions (even though most of the models use versions of GFED). The atmospheric inversion models in MIP optimize land and ocean fluxes, i.e. fire and vegetation fluxes are summed up for the optimization and only a joined 'land' posteriori flux (= NBP) is estimated and provided. Therefore, differences in the assumed fire fluxes only cause the land prior flux to differ among the models (This is the case anyways as the models use different vegetation flux priors). We do not use the MIP fire fluxes and vegetation fluxes separately from each other, but only the joined land estimate. Therefore, the different fire fluxes do not impact our results.*

- Line 155: Might be useful to indicate how many 1x1 degree boxes fall within the study domain.

  *We will mention this in the caption of Figure 1. See new Figure 1 above.*

- Subsection 2.3.2 Bottom-up: seems to me like a lot of magic happens in here. I'm having a little bit of trouble following it, as well as the implications of some of the assumptions that are made. Specifically;
  - what is the purpose of Eq. 2? The text states that most of the TRENDY models provide NBP, GPP, RA, RH. But for the few that don't, an expression is needed to derive the NBP. But I'm having trouble getting Eq. 2 to fall out:

    $$NBP = NEE + fire + flux \ (1)$$

Let NEE=GPP-R. That's the most basic equation, right?
Expanding, gives: GPP-RH-RA. IF NPP=GPP-RA (as stated in the text), then

$$NBP = NPP - RH + fire + flux \quad (2)$$

In my version the sign is backwards compared to Eq 2 in the paper, i.e., NPP-RH versus RH-NPP?

*We included Eq. 2 to clarify*

1) *the sign convention in our study i.e. for which parameter positive fluxes denote a release into the atmosphere*
2) *how we combine the Trendy models' parameter.*

*NEE is defined to be positive for $CO_2$ emissions from the land into the atmosphere i.e. NEE = R – GPP (see for example the definitions in table 1 in Keenan and Williams, 2018: https://doi.org/10.1146/annurev-environ-102017-030204). Therefore:*

*NEE = RH + RA – GPP = RH – NPP*

*So that*

*NBP = RH – NPP + fire + fluc*

*(NPP is defined inversely: positive fluxes denote an uptake of $CO_2$ into land)*

- A couple of the TRENDY models do not provide NBP directly, so it must be calculated from RH - NPP, but without fire and fluc. That seems like a pretty unfair comparison? Please explain.

  *Yes, that's an unsatisfactory circumstance. The TRENDY models largely differ in the fire and fluc fluxes they model. Some models provide fire fluxes with seasonal variability, others assume annually constant fire fluxes, and some models don't consider fire fluxes at all (see Table 1 and text in Bastos et al. (2020) and Table A1 in Le Quéré (2018)). That means that some models (e.g. IBIS) report their NBP data as NBP = R – GPP. For this reason, we decided to handle DLEM and CABLE-POP, which do not provide NBP, fire and fluc fluxes, the same.*

  *In the new TRENDYv11 product DLEM and CABLE-POP report NPB estimates. In TRENDYv11, DLEM NBP equals the difference of DLEM total respiration and DLEM GPP, so that our handling of DLEM is fully correct. For CABLE-POP in TRENDYv11 we find a difference between NBP and R – GPP, caused by fire and fluc fluxes (both not provided). However, the differences are small. The mean monthly difference is smaller than 1% of the seasonal NBP flux' amplitude, so that there is no impact on our results regarding the model selection and flux variability analyses.*

*It is important to note that we do not only use the NBP estimates, but we repeat all the analyses, most importantly the model selection (see Figure 7 c and d), also with NEE. Doing so, we try to exclude the uncertainties introduced by the fire and land-use change fluxes.*

- Are "fluc" completely ignored, being considered insignificant? Or too unknown or too complicated to deal with?
  *The fluc fluxes are partly included in the used estimates. They are included in the atmospheric inversion (TM5-4DVar/GOSAT+IS, in-situ-based inversions) NBP estimate, as it estimates the net flux exchange between land and atmosphere including fluxes due to land use changes. In TRENDY, the definition of NBP explicitly includes fluc fluxes according to the TRENDY protocol (http://blogs.exeter.ac.uk/trendy/files/2022/12/trendy_listofvariables_GCP2022.xlsx), even though the individual models handle fluc differently.*
  *Fluc are not considered in FLUXCOM NBP, which we calculate as FLUXCOM NEE + GFED fire. Furthermore, fluc is not taken into account in the satellite based NEE estimate (TM5-4DVar/GOSAT+IS - GFED fire). Using the TRENDY models fluc estimates (calculated as NBP − NEE - fire), mean fluc fluxes are ~4% of monthly NBP and less than 1% of the monthly GPP, RH, and RA. The IAV variability of fluc is ~5% of the NBP IAV. The land use change fluxes, as given by TRENDY, therefore do not contribute significantly to seasonal nor inter annual variability.*

- Section 2.4 Other datasets
  - Please specify why the GOME-2 product was used rather than a GOSAT SIF product?
    *GOME-2 has much better spatial and temporal data sampling compared to GOSAT and therefore allows continues spatial mapping (Joiner et al., 2013, Köhler et al., 2015). As we want to compare SIF against monthly GPP provided by TRENDY for our whole study region, we use GOME-2 data.*
  - Line 167-168: One of my major comments was about the assumption of linearity in GPP and SIF. A quick google scholar search showed that this seems to still be a debate in the literature? Example papers: [Pickering, EGU Biogeosciences, 2022], [Pierrat, JGR Biogeosciences, 2022], [Zheng, 2024]

*Several studies found SIF to be highly correlated with GPP. Non-linearity occurring on local und sub-daily scale due to GPP light saturation effects (Pierrat et al., 2022) were found to vanish when aggregating SIF on a monthly (Pierrat et al., 2022, Zhang et al., 2016A) or biome scale resolution (Pickering et al., 2022, Zhang et al., 2016B). However, the slope of the linear relationship between SIF and GPP differs between biomes (Sun et al., 2018, Guanter et al., 2012, Liu et al., 2017).*

*In our analyses and model selection process we tried to avoid uncertainties associated with assuming a particular quantitative relationship between SIF and GPP. Therefore, we only used the seasonal timing of SIF as a criterion by normalizing SIF and TRENDY GPP. Doing so, we only assume increasing GPP with increasing SIF (not even necessarily linearly), which is not under debate. Only strongly different SIF seasonal cycles for the different biomes could conflict with our method.*

*To rule out this potential problem, we analyzed GOME-2 SIF individually for the different biomes (see Fig. RC1_1 below). It becomes clear that SIF in the dominant biomes (savanna and grasslands) has the same seasonal cycle and timing of minimum and maximum SIF. SIF in the barren regions has a slightly later minimum in SIF. However, due to the sparse vegetation, the SIF signal and the expected GPP signal in this region is small and cannot dominate the seasonality.*

[Figure]

*Figure RC1_1: Seasonal cycle of SIF for different biomes. The mean seasonal cycle of GOME-2 SIF over the period 2009-01/2018 in the biomes savanna and forest, grasslands, and shrub and barren (according to the land-cover classification in Fig. 1) are shown.*

- Section 3.1 Monthly $CO_2$ concentrations by atmospheric inversions
  - Fig. 2: The spread in red indicates difference between RemoTec and ACOS. They tend to agree well for the uptake season, but tend to diverge more widely for the emissions season. Is there any hypothesized reasons for this? This relates back to the comment about showing a bit of analysis comparing RemoTec and ACOS $XCO_2$ earlier in the paper. (but maybe that is too much detail for the current work!).

*Taking also into account the comments of reviewer 2 we will add a figure showing the mean monthly concentrations of the soundings included in ACOS and RemoTeC to the appendix (see below). There are 37563 soundings in the intersection (compared to 69708 ACOS and 47989 RemoTeC soundings). The monthly concentrations of the intersection follow the unfiltered ACOS and RemoTeC datasets closely with deviations in sub-ppm scale. Therefore, different sampling of ACOS and RemoTeC measurements does not explain the differences between the two datasets.*

*Both, ACOS and RemoTeC, apply a bias-correction after initially retrieving the $CO_2$ concentrations from the measured spectra. Thereby, RemoTeC does only use a global factor to correct the raw $XCO_2$ data over land. ACOS uses several correction terms including dependencies on aerosol optical depth and albedo structures. The differences in the raw $CO_2$ concentrations' seasonal cycle between ACOS and RemoTeC is even larger but reduces with the ACOS bias-correction. Therefore, fundamental methodological differences between the ACOS and RemoTeC retrieval might cause the deviations. Besides the mentioned differences in the applied bias-correction, ACOS and RemoTeC differ in the handling of surface pressure and microphysical aerosol properties. This might contribute to the observed differences. A more detailed analysis would be very interesting and insightful but is beyond the scope of this paper.*

[Figure]

Figure A1: Monthly southern African detrended $CO_2$ concentrations measured by GOSAT. GOSAT/ACOS is given in black, GOSAT/RemoTeC is given in red. Dashed lines show the mean $CO_2$ concentrations over the whole dataset. The mean $CO_2$ concentrations of the soundings included in both datasets, ACOS and RemoTeC, are given as solid line. CS stands for co-sampled and indicates that only soundings, also included in the other dataset are considered. The deviations due to different sampling are in sub-ppm scale and do not explain the differences between ACOS and RemoTeC. Modelled posterior $CO_2$ concentrations of the in-situ-only inversions are co-sampled (cs) on GOSAT and depicted as mean in blue for comparison. The shading indicates the range among the individual in-situ-only inversions. Panel (b) shows the mean seasonal cycle 2009–2018 with the standard deviation over the years as shading.

- Section 3.2 Southern Africa top-down and bottom-up $CO_2$ fluxes
  - Line 229: It seems like the analysis of the individual models could be given in a bit more detail. I'm not sure if it would fit here or in the Appendix around Fig A3.
    *We will add two new figures about the MIP models in the appendix. Furthermore, we will add a discussion paragraph in the figure caption of Fig. A3 (new Fig. A7). Please find the additional figures and discussion text in the answer to your major comment, above.*
  - As mentioned in the Major comments, it seems like a potential stronger conclusion is that the satellite observations do indeed well inform the atmospheric inversions models for flux estimates. Or is that considered to be too "old news"? Might be good to talk it up a bit.
    *We will add the following sentence in the conclusion of section 3.2 ("top-down and bottom up fluxes"):*
    *"Our results support current studies (e.g. Basu et al., 2013; Sellers et al., 2018; He et al., 2023) reporting that satellite observations do well inform atmospheric inversions for flux estimates on sub-continental scales."*
  - Why do the other models (the ones that dont represent well the TM5-GOSAT inversions) stick closer to the prior? Is it simply a matter of loosening the prior covariances in those models?

*We cannot answer these questions with the data provided in the MIP dataset (prior and posterior fluxes, OCO-2 co-samples). To investigate these questions, at least the assumed prior and measurement errors and covariance data is needed. Furthermore, sensitivity tests like model runs with a common prior flux assumption and common error assumptions could help.*

- Fig. 5: GOSAT generally peaks higher than MIP/OCO-2. Seems like it might be useful to directly compare MIP/TM5-4DVar OCO-2 vs TM5-4DVar GOSAT for the overlapping time period?

  *We will include MIP/TM5-4DVar in Fig. 5 and discuss the differences in the main text:*

  *"When directly comparing the two TM5-4DVar inversions (TM5-4DVar/GOSAT+IS and TM5-4DVar/OCO-2+IS, Fig. 5), the latter has smaller emissions. This is most likely a result of the slightly smaller seasonal amplitude of the $CO_2$ concentrations measured by OCO-2 compared to GOSAT (see Fig. A3)."*

- Section 3.3 GOSAT and SIF atmospheric constraints on TRENDY models
  - Lines 260-269: Two selection criteria are described as (a) agreement between TRENDY NBP and NEE fluxes with TM5-4DVar/GOSAT+IS and , and (b) agreement between TRENDY GPP and GOME SIF. Which models were excluded by which criteria? What can be learned from these differences? Is this expected or surprising behavior (that so many TRENDY models disagree with the satellite derived fluxes).

    *We will rewrite parts of this paragraph to clarify the selection process. Moreover, we will add the following explanation of the implications the selection results have:*

    *"All other models, except for the model OCN, already were excluded in the first step of NBP/NEE comparison. OCN performs well in the NBP/NEE comparison, but shows larger deviations in the SIF/GPP comparison (see Fig. 6). Therefore, it was excluded in the second selection step and is not included in the TRENDY selection. The exclusion of OCN underlines the importance of the SIF/GPP selection and demonstrates that a correct timing of the net $CO_2$ exchange fluxes does not necessarily imply the correctness of the modelled gross fluxes. In general, it is noteworthy that only three out of 18 TRENDY models pass our selection process. This again reveals the large uncertainties associated with the TRENDY ensemble estimate for semi-arid southern hemispheric Africa."*

  - Lines 263-265: What if GPP and SIF are not perfectly linear for these biome types. Is there detailed evidence that it is? How could it affect your TRENDY model selection if the two were non-linear?

*Please see our answer around Fig. RC1_1 to your comment about "Section 2.4 Other datasets" above.*

- Section 3.4 Seasonal and IAV of TRENDY gross fluxes
  - ○ Line 322: "…fluxes into the gross…"
    *We will adapt that.*
  - ○ L324: Replace "RH" with "Heterotrophic respiration" at the beginning of a sentence.
    *We will adapt that.*
  - ○ Fig A7: I guess I don't understand why this figure was relegated to the Appendix!
    *The mean seasonal cycle of net and gross fluxes given in Fig. 8 in the main text is clearer and sufficient to support our main conclusions. Therefore, we decided to move the detailed time series of the monthly net and gross fluxes to the appendix.*
  - ○ Line 336: missing "the" before "Birch effect".
    *We will adapt that.*
  - ○ Line 344: Restructure sentence to "A prolonged emission phase of an additional 1 to 2 months…"
    *We will adapt that.*
  - ○ Line 346: The conclusion "enhanced soil respiration due to the drier conditions" seems to be in direct contrast to the statement on L334! Was it just a typoe to replace "drier" with "wetter" here?

    *No, it is no typo. Drier conditions during the year before the rainy season starts can lead to less soil respiration during the year and an enhanced accumulation of soil carbon. With the beginning of the rainy season more soil carbon is available and can be respired once increasing soil moisture allows soil respiration.*

    *We observed in Australia that during wet years nearly no respiration pulses and therefore no time lag between rh and GPP takes place as there are less pronounced rewetting conditions. In dry years with a stronger rewetting, a larger dephasing can be observed. We tried to clarify that in the main text:*

    "…enhanced soil respiration due to the drier conditions causing an enhanced accumulation of soil carbon during the years"

- Section 4 Conclusions
  - ○ L365: This is a nice general conclusion that the satellite observations do in fact provide useful information to the inversion models. Is it fair to make a general statement to the effect that likely some inversion

models are providing too much constraint on the priors such that the satellite observations are not being used to their full potential?

*Our MIP analyses do not allow this general statement. Our findings are consistent with the hypothesis that loosening the prior constraints would lead to a better agreement of MIP with TM5-4DVar/GOSAT but there is not sufficient proof that. To test such a hypothesis, more ancillary, currently not provided, MIP data (like the used error covariance matrices) and sensitivity tests (e.g. same prior fluxes, assumed errors for all models) would be needed.*

o L369-370: I guess I have a little bit of confusion as to how novel the conclusion is that "IAV in southern Africa is driven by GPP variability" since L361-362 indicate that the result was already published? Please clarify.

We will adapt the sentences as follows:

"Using the satellite based selection of TRENDY DGVMs, we find that IAV of NBP and NEE in southern Africa is driven by GPP variability. This supports findings by Ciais et al. (2009), Weber et al. (2009), and Williams et al. (2008) using individual vegetation models"

o Line 375: The conclusion about the importance of properly representing the response of semi-arid regions to soil rewetting in DGVMs seems here to be a little bit of a side-note. That is, the implications or next steps of the conclusion are not given. How is that response parameterized within DGVMs? Is it highly variable model to model? Is the physics completely missing in some models, or just needs to be tuned? Maybe here make the explicitly comment as to what happens when DGVMs misrepresent the soil rewetting. The modelers want/need to know what they can do to improve. According to L35-36 you also feel that is important!

o *We will add a detailed discussion of the soil rewetting implementation in Section 3.4. Please find the discussion text in the response to one of your first minor comments about Lines 58-60.*
*Furthermore, we will add the following at the end of the conclusion:*
"Our results emphasize the importance of correctly representing the response of semi-arid ecosystems to soil rewetting in DGVMs (e.g. different response times of RH and GPP), as this was found to be a prerequisite to accurately capture the seasonal carbon cycle dynamics."

o Is it a fair statement that seasonal variability is driven to first order by precipitation? And IAV is driven by GPP, but what drives that? It's not related to fires?

*We do not state that the seasonal variability is mainly driven by precipitation. In our study we identify one of the key processes shaping*

*the seasonal variability to be the dephasing of heterotrophic respiration and plant $CO_2$ fluxes (GPP-RA). The dephasing is driven by the onset of the rainy season. However, regarding the role of the dephasing, we use the wording 'the seasonal cycle is substantially influenced by enhanced soil respiration' (abstract) and 'the seasonal variability […] is impacted by soil respiration dynamics'.*

*We will include a small discussion about the driver of GPP IAV. To do so, we will add the annual GFED fire fluxes to Fig. A4 (new Fig. A10, see below) and will include the precipitation, soil moisture and temperature anomalies for the subregions in Fig. A5 (now Fig. A11). Furthermore, we will include the following sentences in the discussion of GPP IAV at the end of section 3.4.*

"Looking at the subregions (Panels (d) and (f)), one can see that the sinks in 2010-2012, 2011 and 2017 are mainly driven by the southern grassland region, where enhanced precipitation occurred during these years (see Fig. A5). whereas the comparably large release in 2016 seems to be driven by the whole African region experiencing the highest annual temperatures and driest conditions within the 10-year study period. Therefore, GPP IAV seems to be heavily impacted by precipitation variability. According to GFED (see Fig. A10), fire emissions play a minor role in driving GPP and NBP anomalies. The variability of fire emissions is much lower than for NBP and GPP-RA. In the whole study region, IAV (calculated as standard deviation over the years) of GPP-RA and NBP fluxes are 97.7 TgC/year and 94.1 TgC/year, respectively. IAV of GFED fire emissions is 27.3 TgC/year, similarly low as IAV of RH (27.1 TgC/year). Furthermore, the annual fire emissions do not amplify the trend of the NBP anomalies. They have been on a normal level during the large positive NBP anomaly in 2016. Higher than average fire emissions counteract the sink anomalies in 2011 – 2012 and only the slightly reduced fires in 2017 amplify the sink anomaly."

[Figure]

Figure A10: $CO_2$ fire emissions in southern Africa. The monthly $CO_2$ fire emissions collected by three fire emission databases (GFED in orange, Global Fire Assimilation System (GFAS, Kaiser et

al., 2012) in red and the Fire INventory from NCAR (FINN ,Wiedinmyer et al., 2011) in purple). Furthermore, the annual (July - June) GFED fire emissions are given with the right y-axis. Please note, that the right y-axis starts at 280 TgC per year for better visualization of the fire emissions.

- o Fires are a huge driver from year to year based on precip and temperature extremes. You talk about fires in Sec 3.3 Line 280-289. It seems like the conclusion that "fire fluxes in the DGVMs do not agree to the GFED fire fluxes" should also be mentioned explicitly in the abstract and conclusions sections?

  *We do not evaluate the accuracy of fire fluxes in DGVMs. Furthermore, we make our results as independent as possible from the DGVMs fire flux assumptions by also using the vegetation models' NEE estimates next to the NBP estimates. We prefer not to include a statement about the fire representation in DGVMs in the abstract or conclusion. This would highlight a side-note finding disproportionally compared to our main findings.*

- Appendix A
  - o Fig A1: It is difficult to distinguish the blue from the black color.

    *We will move OCO-2 depicted in black to a new figure, like suggested in the next comment, to make the comparison clearer (see figure below).*

  - o Fig A1: It might be useful to have a version of the plot showing just GOSAT and OCO-2 for 2014 - 2019 (overlap) time range. It could be 2 panels, with the lower panel showing the delta versus time.

    *We will include the following new figure showing GOSAT and OCO-2 only for the years 2015-2018. We increased the linewidth of OCO-2 to make it easier to distinguish.*

[Figure]

Figure A3: Monthly southern African detrended $CO_2$ concentrations given by inversions and satellites. Like Fig. 1, but with detrended $XCO_2$ measurements of OCO-2 in black. Panel (a) gives the monthly mean $CO_2$ concentrations, while Panel (b) shows the mean seasonal cycle 2015-2018. The shading indicates the range among GOSAT/ACOS and GOSAT/RemoTeC and the range among the three in-situ-only

inversions in Panel (a). In Panel (b) the shading indicates the standard deviation over the years.

- o Fig A2: Why is this figure relagated to the Appendix while Fig 5 is in the main body?

  *The focus of our study is on the evaluation of the $CO_2$ flux dynamics in southern Africa. Furthermore, the significance of the comparison to COCCON is limited as we cannot perform a co-sampling (see two comments further down). However, in the comparison of TM5-4DVar/GOSAT+IS against MIP/OCO-2 we can quantitatively compare the 'same' quantity (fluxes over the whole region) even for a larger period of time.*

- o Fig A2: Might be useful if this figure ranged only 2017-2018 to highlight the comparison against COCCON.

  *Thank you for the suggestion. We will adapt the figure as suggested:*

[Figure]

Figure A4: Monthly southern African detrended $CO_2$ concentrations given by inversions, satellites and COCCON measurements. Like Fig. 1, but only for 01/2017-02/2018 and with detrended $XCO_2$ measurements of the COCCON stations Gobabeb in black. The full dataset of COCCON measurements is used, without performing a co-sampling on GOSAT measurements nor further filtering.

- o Fig A2: Might be useful to add a second panel showing the delta XCO2 to highlight differences.

  *The COCCON measurements are not co-sampled on GOSAT (see also the added clarifying last sentence in the caption above). We only use the data to qualitatively point out that the prolonged $CO_2$ concentration maximum in the seasonal cycle is seen by a local station. However, we hesitate to compare the GOSAT and COCCON $XCO_2$ measurements quantitatively by calculating their differences as they measure different quantities. GOSAT measures the $CO_2$ concentrations over the whole*

*study region, whereas COCCON measures locally in the Namibian desert (and direct coincidences are too sparse to evaluate).*

- o Fig A3: I'm having a difficult time interpreting this figure. I don't necessarily see why these 3 models "reproduce the OCO-2 measurements the best" (L230).

  *We will add a second panel to this figure showing the mean flux and XCO$_2$ differences over September – November (see new Figure A7 in the response to your major comment). Doing so, it becomes clearer that the three models which agree well with TM5-4DVar/GOSAT+IS (smallest GOSAT flux mismatch) also agree best with the OCO-2 measurements (smallest OCO-2 XCO$_2$ mismatch).*

- o Fig A3: GOSAT XCO2 is always higher than OCO-2. TM5 GOSAT always gives a greater flux than OCO-2 MIP. This is seen in Fig A2. These are just my observations of interest.

  *That's true! We will include TM5-4DVar/OCO2+IS in Fig. 5. When mentioning this in the text, we also address the differences between OCO-2 and GOSAT:*

  "Still, their estimated emissions are slightly lower than those of TM5-4DVar/GOSAT+IS. When directly comparing the two TM5-4DVar inversions TM5-4DVar/GOSAT+IS and TM5-4DVar/OCO-2+IS (Fig. 5), the latter has smaller emissions. This is most likely a result of the slightly smaller seasonal amplitude of the CO$_2$ concentrations measured by OCO-2 compared to GOSAT (see Fig. A3)."

- o Fig A7: In the legend, does "soil respiration" correspond to what has been termed heterotrophic respiration (RH) throughout the manuscript?

  *Thank you very much for pointing out this mistake! We will correct this in the revised figure.*

- Citations: L688-690: So it looks like OCO-2 v11 data was used for the CO2 concentrations part of the analysis? Is the year 2020 correct? I think it should be 2022. And looks like your DOI is not right (probably for the v10 data?)

  Try this citation: OCO-2/OCO-3 Science Team, Vivienne Payne, Abhishek Chatterjee (2022), OCO-2 Level 2 bias-corrected XCO2 and other select fields from the full-physics retrieval aggregated as daily files, Retrospective processing V11.1r, Greenbelt, MD, USA, Goddard Earth Sciences Data and Information Services Center (GES DISC), Accessed: [Data Access Date], 10.5067/8E4VLCK16O6Q

  *Thank you for this correction. Yes, we are using V11.1r (OCO2_L2_Lite_FP)*

**4 Citations**

1. Jacobs, AMT, 2024, https://doi.org/10.5194/amt-17-1375-2024

2. Pandy, AGU Advances, 2024, https://doi.org/10.1029/2023AV001145

3. Pickering, EGU Biogeosciences, 2022, https://doi.org/10.5194/bg-19-4833-2022

4. Pierrat, JGR Biogeosciences, 2022, https://doi.org/10.1029/2021JG006588

5. Taylor, ESSD, 2022, https://doi.org/10.5194/essd-14-325-2022

6. Taylor, AMT, 2023, https://doi.org/10.5194/amt-16-3173-2023

7. Zheng, Inter. Jour. Applied Earth Observation and Geoinformation, https://doi.org/10.1016/j.jag.2024.103821


14 Oct 2024

-

**Review Summary**

The study by Metz et al. starts from satellite-retrieved XCO2 data analysis over a large southern African region dominated by semi-arid savannas and grasslands. They find a larger seasonal variability of XCO2 from GOSAT and OCO-2 compared to an ensemble of in-situ-optimized inverse models resampled at GOSAT soundings. They further point out that the discrepancy originates from the biospheric prior fluxes that control the optimized biospheric fluxes due to the lack of observational constraints. An understanding of the processes that affect the CO2 fluxes and their variability is gained through a comparison of the TRENDY v9 ensemble of Dynamic Global Vegetation Models (DGVMs).

The paper is well in the scope of Biogeosciences. The paper points out large discrepancies in DGVMs over semi-arid regions in southern Africa, and presents how satellite data can be used to evaluate these models. While the results of the paper do not include completely novel findings (like the authors already point out with appropriate references in the paper), the results are in agreement with previous studies carried out using other approaches, and should therefore be interesting to the readers of the journal. The paper is particularly well written, and the methods and results are clearly presented. I also like how the paper advances from one result to the next one in a way that is pleasant to follow. I recommend publishing the paper in Biogeosciences after the authors have considered the constructive comments detailed below. I have also gone through the comments from Reviewer 1 and try not to repeat their message, although some of my comments might be of similar nature.

**Major comments**

1)

I don't think the title of the paper is fully descriptive of its contents. Could the title be for instance along these lines: "The potential of CO2 satellite observations to inform and evaluate modelled carbon dioxide fluxes in Southern Africa" or something similar. At least, I would propose to change "carbon" to "carbon dioxide" or "CO2" because methane and CO are not discussed in the paper.

*We tried to find a universal title as our manuscript has multiple facets including methodological findings (as described by your suggested title) but also findings about processes driving the African carbon flux variability. Therefore, we would like to keep our original title. We will change the 'carbon' to 'CO₂'.*

2)

The study region is very large and the CO2 fluxes can be quite heterogenous within the study region (for example, on NOAA's website one can see that CarbonTracker concludes part of the study region to be a net source and part a net sink of CO2; https://gml.noaa.gov/ccgg/carbontracker/fluxmaps.php?region=afr&average=annual#imagetable and even though CT is not the best-performing model in the region as the authors point out, this underlines the heterogeneity of the region and makes one wonder how much you can average over without losing important information). I think that the averaged XCO2 time series (Fig. 2) is to some extent dependent on the sampling. The number of GOSAT observations likely varies from only few tens to some 1000-2000 per month over this region, depending on the season. It would be good to show the time series of the number of GOSAT observations (both products) for example added in Fig. 2 and also a map of the spatial sampling density for example around the XCO2 minimum and XCO2 maximum (this could be in the Appendix), and discuss how the sampling affects the results of the concentration-based analysis and the inverse modelling results.

*First of all, it is important to note that the atmospheric inversion systems use single-sounding CO₂ concentration measurements. There is no averaging over the whole month or the whole region done, before feeding the satellite measurements into the inversion system. By modelling the atmospheric transport, the inversion system accounts for the sampling of the satellite data. So there is no information loss due to averaging in the inversion. In case of low measurement numbers, the inversion systems tend to rely more on the used prior assumptions while taking into account the prior error and measurement error assumptions.*

*We show the mean monthly CO₂ concentrations mainly for an illustrative purpose. Doing so, we want to give a first impression of the seasonal dynamics of the satellite measurements. Furthermore, we already want to point out that there are inconsistencies between satellite measurements and (co-sampled) in situ measurement informed inversions, so that satellite measurements are a promising additional information source. However, the rest of the manuscript and analyses do not rely on the concentration analysis. We will add the following sentence to the methods to clarify that:*

*"To this end, we use the model TM5-4DVar and assimilate GOSAT CO₂ concentration measurements over land and ocean together with the in situ measurements. We use the individual total CO₂ concentration measurements, i.e. we do not apply any detrending or spatiotemporal averaging. Detrending and spatiotemporal averaging is only applied for*

visualization purposes to show the variability in the monthly $CO_2$ concentrations (Section 3.1)."

*Furthermore, we will discuss the satellite data abundance. Taking into account the suggestions of you and reviewer 1, we will add the following two figures in the appendix and sentences in the main text:*

[Figure]

Figure A6: Number of satellite measurements per month. The amount of satellite measurements of the GOSAT/ACOS (red dashed), GOSAT/RemoTeC (dark red solid), and OCO-2 (grey dotted) dataset are given. Note that the number of OCO-2 measurements is shown divided by 100 to enable a comparison to the much less abundant GOSAT measurements.

[Figure]

Figure A5: Number and distribution of satellite $CO_2$ concentration data above southern Africa. (a), (d), and (g) Total number of GOSAT/ACOS, (b), (e), and (h) GOSAT/RemoTeC, and (c), (f), and (i) OCO-2 data per 3°x2° grid cell for (a) - (c) the months of carbon uptake (January – June), (d) - (f) the emission season (July – December),

and (g) – (i) the months with the strongest emissions. GOSAT/ACOS and GOSAT/RemoTeC measurements from 2009 to 2018 and OCO-2 measurements from 09/2014 to 2018 are included. The maximum of the color scale is the same for all time periods, but different for OCO-2 than for GOSAT/ACOS and GOSAT/RemoTeC. Compared to GOSAT/ACOS, GOSAT/RemoTeC has a reduced number of measurements, as RemoTeC applies stricter filtering of the GOSAT soundings.

We will add at the end of Section 3.1. ($CO_2$ concentration results):

"The number of GOSAT measurements (see Fig. A5 and Fig. A6) is variable throughout the year with the smallest numbers occurring during the rainy season around December and January. This leads to larger uncertainties in the monthly mean satellite $CO_2$ concentrations and satellite-based fluxes during the transition from maximum to minimum concentrations and fluxes."

*The impact of sampling on the differences between GOSAT/ACOS and GOSAT/RemoTeC is further discussed below with one of your minor comments.*

**Minor comments**

Abstract: I suggest highlighting the discovered role of grassland carbon uptake for IAV which I consider as a key result of the work. It would also be helpful to give key numbers of TgC so that the reader can contrast those more easily.

*We will mention grasslands explicitly in the abstract. Also taking into account the comments of reviewer 1, we will adapt the sentence as follows:*

"Doing so, our satellite-based process analyses pinpoint photosynthetic uptake in the southern grasslands to be the main driver of inter-annual variability of the southern African carbon fluxes, agreeing with former studies based on vegetation models alone."

*We will furthermore add key numbers of IAV in the discussions of the main driver of NBP IAV:*

"According to GFED (see Fig. A10), fire emissions play a minor role in impacting GPP and driving NBP anomalies. The variability of fire emissions is much lower than for NBP and GPP-RA. In the whole study region, IAV (calculated as standard deviation over the years) of GPP-RA and NBP fluxes are 97.7 TgC/year and 94.1 TgC/year, respectively. IAV of GFED fire emissions is 27.3 TgC/year similarly low as IAV of RH (27.1 TgC/year)."

Line 46: Spell Orbiting Carbon Observatory -2 with a hyphen

*Thank you for the corrections. We will check and correct this and the following four mistakes.*

Lines 56-57: CO2 exchange fluxes → CO2 exchange

Line 57: to flux estimates → to those (to avoid repetition)

Line 69: Valentinti → Valentini (check!)

Line 69: grass land → grassland

Figure 1: This is a massive study region. To make this clearer to the reader, I suggest adding a scale bar representing for example 200 or 500 km or similar. In addition, please add the locations of the Gobabeb COCCON and the Kruger National Park measurement with pins or equivalent symbols.

*We will do that (please find the updated figure in the response to reviewer 1).*

Line 76: I believe that the description of the timing of the fire season applies to the study region in question and not the whole Africa. Please specify in the text.

*We will specify this in the text as follows:*

"The fire season starts in May in the western part of southern hemispheric Africa and spreads …"

Section 2.2: Did you also include satellite data over the ocean in your analysis?

*No, for the analyses in Section 2.2 we did not use measurements over the ocean. We will specify this in a revised manuscript:*

"… measured by the Greenhouse Gases Observing Satellite (GOSAT**) over land in our study region**."

Line 82: column-average → column-averaged

*We will adapt that, thank you.*

Line 85: How different is RemoTeC v2.4.0 compared to RemoTeC v2.3.8? Please summarise the key developments after v2.3.8 in the text.

*We will include*: "The major updates between versions 2.3.8 and 2.4.0 are stricter quality filtering in the latter and updated ancillary input data, especially for the prior gas concentrations used."

Lines 87-90: I understand that both products have been used as two separate products (e.g., apply their own quality filtering and potential bias corrections). Did you compare ACOS and RemoTeC by considering only the exact same GOSAT soundings? If so, what did you learn about the product XCO2 differences? While this question may not be in the core of the paper, I think it is of high interest to learn as much as possible about the satellite observations' differences in particular over these regions that are poorly sampled by any other measurements. In addition, the differences in the satellite products can result in flux differences of several tens TgC per month for this region (Fig. 3), which is significant and interesting. Depending on the authors' view, this might even merit a sentence in the Conclusions to address the need of validation measurements in this region to be able to reconcile residual differences in the satellite products.

*We also had a look at the exact same soundings included in GOSAT/ACOS and GOSAT/RemoTeC. There are 37563 soundings in this intersection (compared to 69708 ACOS and 47989 RemoTeC soundings). Please find below the time-series of monthly $CO_2$ concentrations given by the intersection. The monthly $CO_2$ concentrations of the intersection calculated by ACOS and RemoTeC are similar to the $CO_2$ concentrations of the whole ACOS and RemoTeC dataset, respectively. The deviations due to different sampling are in sub-ppm scale and do not explain the differences between ACOS and RemoTeC. We will add this figure to the appendix.*

[Figure]

Figure A1: Monthly southern African detrended $CO_2$ concentrations measured by GOSAT. GOSAT/ACOS is given in black, GOSAT/RemoTeC is given in red. Dashed lines show the mean $CO_2$ concentrations over the whole dataset. The mean $CO_2$ concentrations of the soundings included in both datasets, ACOS and RemoTeC, are given as solid line. CS stands for co-sampled and indicates that only soundings, also included in the other dataset are considered. The deviations due to different sampling are in sub-ppm scale and do not explain the differences between ACOS and RemoTeC. Modelled posterior $CO_2$ concentrations of the in-situ-only inversions are co-sampled (cs) on GOSAT and depicted as mean in blue for comparison. The shading indicates the range among the individual in-situ-only inversions. Panel (b) shows the mean seasonal cycle 2009–2018 with the standard deviation over the years as shading.

Both, ACOS and RemoTeC, apply a bias-correction after initially retrieving the $CO_2$ concentrations from the measured spectra. Thereby, in RemoTeC only a global factor to correct the raw $XCO_2$ data over land is used. ACOS uses several correction terms including dependencies on aerosol optical depth and albedo structures. The differences in the raw $CO_2$ concentrations' seasonal cycle between ACOS and RemoTeC is even larger but reduces with the ACOS bias-correction. Therefore, fundamental methodological differences between the ACOS and RemoTeC retrieval cause the deviations. Besides the mentioned differences in the applied bias-correction, ACOS and RemoTeC differ largely in the handling of surface pressure. This might contribute to the observed differences. As we can only speculate about the reasons for the differences between the monthly ACOS and RemoTeC $CO_2$ concentrations, we will not include this analysis in the revised manuscript.

It is important to mention that the differences between ACOS and RemoTeC (see shading in Fig. 2) are smaller than their difference to and the deviations among the in situ based atmospheric inversions. Furthermore, as we assimilate ACOS and RemoTeC separately from each other in TM5-4DVar, we can estimate and do show (see shading Fig. 3) the uncertainties in the fluxes associated to the retrieval differences. Here again, the differences between TM5-4DVar/RT240 and TM5-4DVar/ACOS are much smaller than the differences to and the deviations among to other datasets (IS-based inversions, FLUXCOM and TRENDY) underlining that satellite $CO_2$ concentration measurements and satellite-based fluxes have a discriminating power and can be used as atmospheric constraints.

*We prefer not to include a statement about the need for further validation measurements in the conclusion. Such a statement is already included in the manuscript at the end of section 3.1. Moreover, even though it is of scientific interest to have more validation measurements to further evaluate the differences between ACOS and RemoTeC, our analyses and conclusions do not rely on that nor would they change when resolving the differences between the retrievals.*

Lines 87-90: In Fig. 2, we are seeing differences in the sub-ppm scale. It would be helpful to give guidance (with references) to the (global) accuracy and precision of these GOSAT products in this part of the text to help the reader understand to what extent we can trust the satellite observations.

*We will add the following sentences in line 90, after introducing GOSAT*:

*"GOSAT/ACOS single measurements have a precision of 1.5 ppm and a mean bias of 0.2 ppm in validation against TCCON (Taylor et al., 2022). GOSAT/RemoTeC was found to have a slightly lower precision of 1.9 ppm (Buchwitz et al., 2017) and by construction a mean bias of 0 ppm in comparison to TCCON after bias correction. GOSAT/RemoTeC was found to have a regional and seasonal systematic error of 0.6 ppm and 0.5 ppm respectively (Buchwitz et al., 2017)."*

Line 91: I think it would be more descriptive to say "evaluation" instead of "validation".

*Agree.*

Line 94: The COCCON data were not co-located with the satellite observations, right? Did you do any filtering to the data, for example in the early or late hours with large solar zenith angles? Please specify in the text.

*No, we did not apply any further filtering. We will include that in the manuscript when we introduce the COCCON data and in the caption of Fig. A2*:

*"We use the full dataset of COCCON measurements i.e. we do not apply further filtering or co-sampling to GOSAT, as there are too few coinciding GOSAT measurements."*

Sect. 2.3.1: The three inversion models also use different data assimilation schemes, right (4DVar vs. enKf)? This could be specified in the text.

*Yes, we will include the following*:

*"TM5-4DVar and CAMS make use of a four-dimensional variational data assimilation, while CarbonTracker uses an ensemble Kalman filter."*

Sect. 2.3.1.: Did you assimilate also GOSAT observations over oceans? Please specify.

*Yes, in our inversion the GOSAT ocean observations are included. We will include: ''... assimilate GOSAT $CO_2$ concentration measurements over land and ocean."*

Sect. 2.3.1.: Question, out of curiosity: did you also consider a GOSAT-only inversion with no in-situ data?

*Yes, we also run a GOSAT only inversion. The GOSAT only inversions (ACOS and RT) show the same seasonal cycle as the GOSAT+IS inversions, but slightly larger amplitudes. We decided to use the GOSAT+IS inversion as it takes into account all measurement information available. Below you find a figure with the comparison.*

[Figure]

Figure RC2_1: Top-down southern African net $CO_2$ fluxes estimated with TM5-4DVar. Net $CO_2$ fluxes estimated by assimilating GOSAT/RemoTeC (red) or GOSAT/ACOS (black) only (dashed line) or in combination with in situ measurements (solid line) are shown.

Line 113 (and elsewhere): please use \citep[TM5-4DVar,][]{Basu et al reference} LaTeX command to get the rid of the extra parentheses within parentheses (if working with LaTeX).

*We will adapt the parentheses in the revised manuscript.*

Lines 141-144: Did you apply the averaging-kernel correction when comparing the cosampled model concentrations to satellite data? The effect is not likely to be large but it might be non-negligible since the differences in question are mostly in the sub-ppm scale.

*In the figures in the manuscript, we did not apply the averaging-kernel (AK) correction. However, we checked its impact for CarbonTracker and CAMS. Below you can find a figure showing the effect of including the AK correction in the comparison to GOSAT. The effect is small and most importantly, does not change the seasonal variability. We calculated the AK effect using the following formula:*

$$XCO2_{AK} = \sum_{h} \left[ \left( CO2_{models,h} * AK_h \right) + \left( CO2_{GOSAT_{Prior,h}} * (1 - AK_h) \right) \right] * p_{w\_h}$$

*Thereby, h stands for the different height layers and p_w is the pressure weight of each layer. The formula is used internally in RemoTeC (CO2_model is then CO2_GOSAT) to*

*account for the AK of the measurements when calculation XCO₂. For CO2_GOSAT_prior an early version of CarbonTracker (v2019) is used. The effect of applying the AK is small and most importantly, does not change the seasonal variability. For this reason, we decided to neglect the AK.*

[Figure]

Figure RC2_2: Monthly southern African detrended $CO_2$ concentrations given by CT2022 and CAMS. The modeled $CO_2$ concentrations are co-sampled (cs) on the locations of the GOSAT measurements (CT2022 dotted, CAMS dash-dotted). In Panel (A) values with averaging-kernel correction (AK) are given in blue, values without the correction in black. Panel (B) gives the differences between co-sampled $CO_2$ concentrations with and without averaging-kernel correction.

Line 156: Does the FLUXCOM have a product version that could be referenced here? We have recently worked with a FLUXCOM product that had a strong positive bias in CO2 fluxes over both Northern and Southern Africa. This seems to not be the case with the product in the paper. In fact, if I interpret Fig. 3 correctly, the FLUXCOM NEE would give out either close to zero or negative values without GFED fire emissions, correct?

*We use FLUXCOM version 1 with the RS_V006 setup. We will add this information in the main text and in the overview of the data in Table A1. The FLUXCOM version agrees with the data used in the overview paper Jung et al. 2020.*

*Yes, FLUXCOM NEE is mainly negative and only slightly positive around September. This agrees with the fluxes for whole southern Africa given in Figure 6 and 7 in the FLUXCOM publication mentioned above (Please note that in Figure 7 the mean annual flux depicted in Figure 6 is subtracted).*

Line 173: Methods → Data and Methods

*We will adapt that.*

Lines 176-178: This reminds me of similar GOSAT-to-model discrepancies that were pointed out by Lindqvist et al. (ACP, 2015; https://acp.copernicus.org/articles/15/13023/2015/) and their reasoning, albeit with much earlier versions of the data and models.

*Thank you for mentioning the paper. Lindqvist et al. (2015) compare the mean seasonal cycle (MSC) of GOSAT/ACOS $CO_2$ concentration measurements to the MSC of TCCON and to the MSC of in situ measurement based atmospheric inversion models. In the latter comparison they find differences between GOSAT/ACOS and the models, which are smaller than the deviations among the models themselves. They conclude that GOSAT/ACOS $CO_2$ concentration MSC may be sufficiently accurate to evaluate land surface models. That matches the methods we use. However, Lindqvist et al. 2015 base their conclusion only on an evaluation of the northern hemisphere. For this reason, we prefer not to cite the paper in our study.*

Figure 2: Please consider adding the monthly data density or, if the result is too crowded, presenting the data density time series using other means.

*We will include the two figures given in our response to your major comment in the appendix of the manuscript.*

Figure 2: What does the CS in the subscript of the "Inverse models" stand for?

*CS stands for 'Co-Sampled on GOSAT'. We will add the abbreviation in the caption of the Figure.*

Line 190: e. g. → e.g.,

*We will adapt that.*

Sect. 3.2: I think it is interesting and worth pointing out how you can already "see" the impact of the fluxes in the detrended concentrations (Fig. 2). The well-aligned timings also suggest that the concentration maxima and minima are driven by the fluxes from this region as opposed to transport from elsewhere. This suggests that satellite data can already in its concentration form have potential and be useful for an evaluation of models in a region (even without inverse modelling!).

*Yes, the timing of the minima and maxima in concentrations is the same for the maximum emissions and maximum uptake. However, disentangling local fluxes and transport effects is very difficult, even after subtracting a global background. The same timing in both does not necessarily imply transport effects to be irrelevant. For this reason, we prefer not to include such an additional statement.*

Figure 3 (a): the legend is partly on top of the plotted data. Please revise such that the data are fully visible. This comment applies to some subsequent figures as well.

*We will adapt the figures.*

Line 204: remove "methods"

*We will correct our manuscript according to this and your following two comments.*

Line 244: a → an

Line 253: processed based → process-based

Sect. 3.3.: I understand that if the focus of the paper is to learn about the processes affecting the large-scale CO2 exchange in this region, it is meaningful to focus on the DGVMs that agree with the GOSAT inversions. However, as also Reviewer 1 points out, it seems like a missed opportunity to not discuss the models that agree the worst. I also wonder how generalizable these findings might be for other semi-arid regions in the world; one good example already published by the authors about the Australian fluxes (Metz et al., 2023). I get the feeling that the authors have already learned much more from individual model performance than what they are writing in the paper. I know it is not a preferred task for anyone to point out flaws in someone else's product but this is sometimes necessary to enhance openness that might further advance science.

*In general, from the perspective of a non-vegetation modeler, it is difficult to identify the processes missing or inaccurately represented in the DGMVs as the models differ largely in the implemented processes. We found the dephasing of vegetation fluxes and heterotrophic respiration to be one of the key processes characterizing the southern African carbon exchange. Focusing on that and also taking into account the comment of reviewer 1, we will add the following in the discussion of the DGVMs gross fluxes*:

"It is noteworthy that large parts of the not selected 'other' TRENDY models miss the dephasing between RH and GPP-RA. Their NBP estimates, therefore, do not agree with the emissions around October found by the satellite inversion. Implementing soil respiration due to rewetting more accurately in those models could improve their agreement with the satellite-based fluxes. Metz et al. (2023) found that the dephasing in the TRENDY models is most likely caused by a different response time of soil respiration and vegetation growth on precipitation e.g. water needs to percolate into the deeper soil layers with plant roots to initiate plant growth, whereas heterotrophic respiration is driven by upper soil layer soil moisture or precipitation. The implementation of such a time lag between heterotrophic respiration and GPP seems to be a necessary but not sufficient prerequisite to accurately capture the seasonal carbon flux variability in semi-arid regions. Our results call for studies on how to implement the response of ecosystems on soil rewetting more accurately to improve the consistency and accuracy of the TRENDY ensemble in semi-arid regions."

Figure 6: The shading mentioned in the caption is not visible (at least in my printed version). The legend covers part of the data, please revise the figure. Why is GOME-2 SIF used instead of GOSAT SIF or OCO-2 SIF? I wonder how different these would be.

*Thank you for making us aware of that. The shading is visible in the digital version. We will make the shading darker, so that it is also visible when printing the paper and we will move the legend.*

*We do not use GOSAT-SIF because the data is much coarser than GOME-2 data as pointed out in Joiner et al. (2013) and Köhler et al. (2015). We also checked OCO-2 SIF. Please find Figure 6 additionally with OCO-2 SIF below (Fig. RC2_3). It agrees well with GOME-2 and would lead to the same results (i.e. TRENDY model selection). We decided to only use GOME-2 in the publication as it nearly covers the whole period 2009-2018 in contrast to OCO-2 which only provides data from late 2014 on.*

[Figure]

Fig. RC2_3: Like Figure 6, but with monthly mean OCO-2 740nm SIF.

Lines 278-279: CAMS cannot be identified from the spread of models in Figs. 2-3.

*We will change the figure reference to Fig A2 (former A1) and Figure A7 (former A3), which show the GOSAT $CO_2$ concentration together with CAMS concentration (Fig. A2) and the difference of the individual MIP models including CAMS to TM5-4DVar/GOSAT fluxes (Fig. A7).*

Lines 283-284: You could add a very recent reference in your discussion of fire emissions in the regions. This one also seems to suggest underestimated fire emissions. Van der Velde et al. (2024): https://agupubs.onlinelibrary.wiley.com/doi/full/10.1029/2023GL106122

*Thank you for pointing out this paper. We will include the citation in line 284.*

Lines 300 and 304: I think the correct spelling is La Niña and El Niño

*Thank you for this correction. We will adapt our revised manuscript accordingly.*

Line 302: "enhanced surface near soil moisture" this sentence might be missing a word?

*We would rephrase the sentence to*: 'enhanced soil moisture near the surface'

Sect. 3.4 title: I'm not sure what is meant by gross fluxes. Could you just say fluxes?

*We use the term gross fluxes for the constituent vegetation fluxes (GPP, respiration). Combined, they result in the net exchange fluxes. This differentiation between net and gross fluxes is used in literature (e.g. Baldocchi et al. 2018, https://doi.org/10.1016/j.agrformet.2017.05.015). By including 'gross' in the title we want to define the section against the discussions of the net flux variability before.*

Line 321: remove "fluxes"

*We will remove the word.*

Line 344: Suggest to rewrite the sentence starting "A by 1-2 months…" as it is not easy to read.

*Going along with suggestions of reviewer 1, we will rephrase the sentence as following: 'A prolonged emission phase of an additional 1 to 2 months ...'*

Lines 359-362: To me, carbon uptake by grasslands dominating the IAV of the entire southern African fluxes sounds like an important finding that should be brought to the Abstract and perhaps even contrasted in magnitude against fire emissions to the reader so that the magnitude is easier to grasp. At least to me this finding highlights the importance of an improved understanding (and modelling) of the sensitivity of grasslands.

*We will adapt the sentences about the IAV in the abstract to also mention grasslands. Doing so, we also take into account the comments of reviewer 1 concerning the novelty of the findings.*

"Doing so, our satellite-based process analyses pinpoint photosynthetic uptake in the southern grasslands to be the main driver of inter-annual variability of the southern African carbon fluxes, agreeing with former studies based on vegetation models alone."

*Furthermore, we will add a discussion about the impact of fire emissions on the IAV of NBP. In the new paragraph, we also compare the IAV of GFED fire emissions to those of the vegetation fluxes:*

"According to GFED (see Fig. A10), fire emissions play a minor role in impacting GPP and driving NBP anomalies. The variability of fire emissions is much lower than for NBP and GPP-RA. In the whole study region, IAV (calculated as standard deviation over the years) of GPP-RA and NBP fluxes are 97.7 TgC/year and 94.1 TgC/year, respectively. IAV of GFED fire emissions is 27.3 TgC/year, similarly low as IAV of RH (27.1 TgC/year). Furthermore, the annual fire emissions do not amplify the trend of the NBP anomalies. They have been on a normal level during the large positive NBP anomaly in 2016. Higher than average fire emissions counteract the sink anomalies in 2011 – 2012 and only the slightly reduced fires in 2017 amplify the sink anomaly."

Lines 359-362: This is an example where the sampling of the satellite data for GOSAT inversions might end up affecting the resulting reasoning (see my Major comment #2).

*Line 359-362 describes that in 2013, 2015, and 2016 the emission phase is by 1-2 months longer than in the other years. The number of satellite measurements is smallest during the end of the emission phase and the beginning of the growing season. Therefore, the fluxes have a higher uncertainty than in the rest of the year. In 2013, 2015, and 2016 the number of measurements is not especially low compared to other years (see Figure A6 above). However, within a more detailed analysis of the prolonged emission phase, it would be necessary to analyze the measurement locations during these months further to make sure that the flux estimates are reliable. Such a detailed analysis is out of the scope of this paper.*

Line 374: rain induces → should this read rain-induced?

*Yes. We will correct that.*

Figure A1 caption: und → and

*We will correct the typo.*

Figure A1 legend: what does "CS" stand for? Also, the legend is covering part of the data, please revise.

*We will add that cs is the abbreviation for co-sampled. Furthermore, we will move the legend. See new figure below:*

[Figure]

Figure A3: it seems that months 10 and 11 are particularly challenging. Could you please add a bit of discussion related to this in the text?

*Taking also into account the questions and comments of reviewer 1 we will include an additional panel to Figure A3 (now A7), two new supplement figures and a new supplement text evaluating the MIP model fluxes in more detail. Doing so, we will also address the differences between the individual months.*

"In general, the GOSAT flux mismatch and the OCO-2 $XCO_2$ mismatch is larger in October and November than in September. This is most likely caused by the prior fluxes in

September being closer to the GOSAT based fluxes than in the other two months (see panel b in Figure A9)."

*Please find the new figures and complete discussion in text A1 in the response to reviewer 1.*

Figure A8: Just a comment: this amount of precipitation is likely to affect the number of satellite observations of the region. Please see my Major comment #2.

*Yes, we will include the connection between the rainy season and the amount of GOSAT data in the main text as mentioned in the response to your major comment:*

"The number of GOSAT measurements (see Fig. A5 and Fig. A6) is variable throughout the year with the smallest numbers occurring during the rainy season around December and January."

Figure A9: Are these measurements assimilated in any of the in situ optimized inversion models discussed in the paper? The time span would match part of the time period discussed in the paper.

*No, these $CO_2$ flux measurements are not included in the ObsPack dataset, which is used in the in situ optimized inversion models. In general, only $CO_2$ concentration (and not $CO_2$ flux) measurements are assimilated in the inversion models. Even though FLUXNET stations are also measuring $CO_2$ concentrations (not shown in Fig. A9 (new A15)) they are usually not included in atmospheric inversion systems.*